# The significance of coastal bathymetry representation for modelling the tidal response to mean sea level rise in the German Bight

Caroline Rasquin[1], Rita Seiffert[1], Benno Wachler[1], Norbert Winkel[1]

[1]Federal Waterways Engineering and Research Institute, Wedeler Landstraße 157, 22559 Hamburg

*Correspondence to*: Caroline Rasquin (caroline.rasquin@baw.de)

**Abstract.** Due to climate change an accelerated mean sea level rise is expected. One key question for the development of adaptation measures is how mean sea level rise affects tidal dynamics in shelf seas such as the North Sea. Owing to its low-lying coastal areas, the German Bight (located in the south-east of the North Sea) will be especially affected. Numerical hydrodynamic models help to understand how mean sea level rise changes tidal dynamics. Models cannot adequately

represent all processes in overall detail. One limiting factor is the resolution of the model grid. In this study we investigate which role the representation of the coastal bathymetry plays when analysing the response of tidal dynamics to mean sea level rise.

Using a shelf model including the whole North Sea and a high-resolution hydrodynamic model of the German Bight we investigate the changes in M2 amplitude due to a mean sea level rise of 0.8 m and 10 m. The shelf model and the German

Bight Model react in different ways. In the simulations with a mean sea level rise of 0.8 m the M2 amplitude in the shelf model generally increases in the region of the German Bight. In contrast, the M2 amplitude in the German Bight Model increases only in some coastal areas and decreases in the northern part of the German Bight. In the simulations with a mean sea level rise of 10 m the M2 amplitude increases in both models with largely similar spatial patterns. In two case studies we adjust the German Bight Model in order to more closely resemble the shelf model. We find that a different resolution of the

bathymetry results in different energy dissipation changes in response to mean sea level rise. Our results show that the resolution of the bathymetry especially in flat intertidal areas plays a crucial role for modelling the impact of mean sea level rise.

## 1 Introduction

During the 20th century and the beginning of the 21th century an increase and acceleration in global mean sea level rise

(MSLR) have been observed. The global mean sea level rose between 1901 and 1990 at an average rate of 1.4 mm/year (IPCC, 2019). Between 1993 and 2015 this value more than doubled with a rate of 3.2 mm/year. Future predictions show a MSLR of 0.55-1.40 m by 2100 (17.-83. % percentile) in the scenario RCP8.5. This increase could exceed several meters during the 22th century. Many coastal areas will be affected by an accelerated MSLR.

Our study focusses on the German Bight which is located in the southeast of the North Sea (see Figure 1 and Figure 2). This part of the North Sea is characterised by low-lying coastal areas, which are, in contrast to steep coastlines, especially vulnerable in a changing climate. MSLR will not only influence mean water levels in themselves, and so be important with regard to coastal protection and especially storm surges, but will also influence tidal dynamics (e.g. the magnitude of different tidal constituents and current velocities) in the North Sea (Ward et al., 2012; Pickering et al., 2012; Wachler et al., submitted) and its adjacent estuaries (Seiffert and Hesser, 2014; Plüß, 2004). Changes in the tidal dynamics have a number of consequences for the Wadden Sea and the estuaries. Altered sediment transport due to a changed ratio of flood to ebb current velocity will lead to sea level rise-induced morphological changes in the Wadden Sea (Dissanayake et al., 2012; Becherer et al., 2018). Due to MSLR the turbidity zone in the estuaries, which depends on the discharge as well as on the tidal conditions, will shift upstream (Kappenberg and Fanger, 2007; Seiffert et al., 2014). Furthermore, salt intrusion into the estuaries will be affected (Seiffert and Hesser, 2014). Thus, future challenges related to MSLR include not only coastal protection issues, but also other aspects such as sediment management in estuaries functioning as access waterways to ports. Some of the largest ports in Europe such as Rotterdam, Hamburg and Antwerp are located in the south-east of the North Sea. Other challenges involve drainage of the hinterland and the protection of the UNESCO World Natural Heritage Site Wadden Sea that provides a unique habitat for flora and fauna. For the development of potential adaptation measures, it is important to understand how MSLR changes tidal dynamics.

Several previous studies have investigated the impact of MSLR on tidal dynamics in the North Sea, especially on the M2 amplitude, which is the most energetic component (e.g. Ward et al., 2012, Pickering et al., 2012, Idier et al., 2017). Some of these studies came to contradictory results. Ward et al. (2012) analysed a MSLR of 2 m with the shelf model KUTM and obtained a decrease of M2 amplitude in the German Bight whereas Pickering et al. (2012) found an increase of M2 amplitude with the same MSLR of 2 m using the shelf model DCSMv5.

Pelling et al. (2013) provide an explanation for these contrasting results. They show that the differences are due to the way of implementing the landward model boundary in the model simulations. In Pickering et al. (2012) the model has a fixed vertical wall at the boundaries whereas in the study of Ward et al. (2012) new cells of the former hinterland are allowed to flood with MSLR. These new cells provide additional shallow areas of high dissipation resulting in a damping effect that counteracts the general decrease of dissipation due to MSLR. In the model allowing new cells to flood less energy reaches the northern German Bight because of the higher dissipation along the Dutch and German coast. In the model with a fixed boundary, more energy remains in the M2 tide with MSLR due to the lack of additional dissipative areas, leading to an increase of M2 amplitude with mean sea level rise. A study by Pelling and Green (2014)with similar model setups using smaller levels of MSLR (up to 1 m) supports the theory of Pelling et al. (2013). They also suggest that higher resolution simulations with up to date and realistic flood defence representations are needed to estimate changes in tidal dynamics due to MSLR. Not only the adequate representation of flood defence but also the correct description of topography in shallow intertidal regions could be important for the estimation of the system's response to MSLR. In this context the question arises whether the resolution of shelf models such as DCSM or KUTM is sufficient to assess reliably the response of tidal

dynamics in the North Sea to MSLR. In particular, shallow areas of high dissipation might be insufficiently represented in the models.

Due to the relatively coarse resolution of shelf models with a cell size of about 2 to 7 km, topographic features such as estuaries or the details of tidal flats and channels in the Wadden Sea cannot be represented in these models. Thus, potentially
important factors such as missing volume in the tidal basins of the estuaries or inadequately resolved topographical structures might lead to imprecise results. The aim of our study is to investigate whether the response of tidal dynamics in the German Bight to MSLR is sensitive to the resolution-dependent simplifications of shelf models. For this purpose we perform hydrodynamic numerical model simulations with different levels of resolution with regard to bathymetric features and the coastline (model domain).

**2. Methods**

In this study we use two different models. The Dutch Continental Shelf Model DCSMv6FM that simulates the tidal dynamics in the entire North Sea and the German Bight Model (GBM), a higher resolved model that covers the German Bight and its estuaries. The GBM uses boundary conditions from the DCSMv6FM and simulates the tidal dynamics in the German Bight on a more detailed level.

**2.1 Shelf Model: DCSMv6FM**

The Dutch Continental Shelf Model DCSMv6FM (Zijl, 2014) is a 2D-hydrodynamical model based on the shallow water equations. It is a further development of the structured Dutch Continental Shelf Model DCSMv6 (Zijl et al., 2013; Zijl et al., 2015) using the new flexible mesh capacities D-Flow FM (Kernkamp et al., 2011). The flexible mesh technique is based on
the classical unstructured grid concept. In contrast to the German Bight Model, the DCSMv6FM does not include subgrid information. The model domain covers the northwest European shelf (Figure 1). In the North Sea the resolution of the model grid is 1.5′ in the east-west direction. The resolution in the north-south direction is 1′. This leads to a grid cell size of 1.9 by 1.9 km. Beyond the shelf the resolution is coarser with a grid size of about 7.4 by 7.4 km.

The bathymetry is based on data from the North-West Shelf Operational Oceanographic System (NOOS, 2002). These data
are supplemented by data from ETOPO2 (National Geophysical Data Center, 2006). During the calibration process using the OPENDA-DUD algorithm (Garcia et al., 2015), the bathymetry was adjusted in some areas to achieve an improved propagation of the tidal wave. The OPENDA-DUD algorithm defines the calibration as an optimization problem. It takes the bathymetry and the bottom friction coefficient as calibration factors. For further information on the calibration of the DCSMv6 we refer to Zijl et al. (2013).
The model includes tide generating forces. At the seaward open boundary the model is forced by tidal constituents. The amplitudes and phase lags of the 22 main diurnal und semi-diurnal constituents are derived by interpolation from the dataset

generated by the GOT00.2 global ocean tide model (Ray, 1999). Sixteen additional partial tides are adopted from FES2012 (Carrère et al., 2013). External surge is forced as an inverse barometer correction based on time and space varying pressure fields. Atmospheric forcing (wind at 10 m and atmospheric surface pressure) is included by use of the reanalysis data COSMO-REA6 (Hans-Ertel-Centre for Weather Research, Bollmeyer et al., 2015). A constant value of 0 mNHN is set for the initial conditions of the water level. The unit mNHN denotes metres above the German datum which is a good approximation of mean sea level. A sufficiently long initialisation time ensures that a dynamical equilibrium is reached before the simulations start.

## 2.2 Regional Model: German Bight Model

The regional German Bight Model covers the German Bight from Terschelling in the Netherlands to Hvide Sande in Denmark (Figure 2). The estuaries of the rivers Elbe, Weser, and Ems are included with their main tributaries up to the tidal weirs.

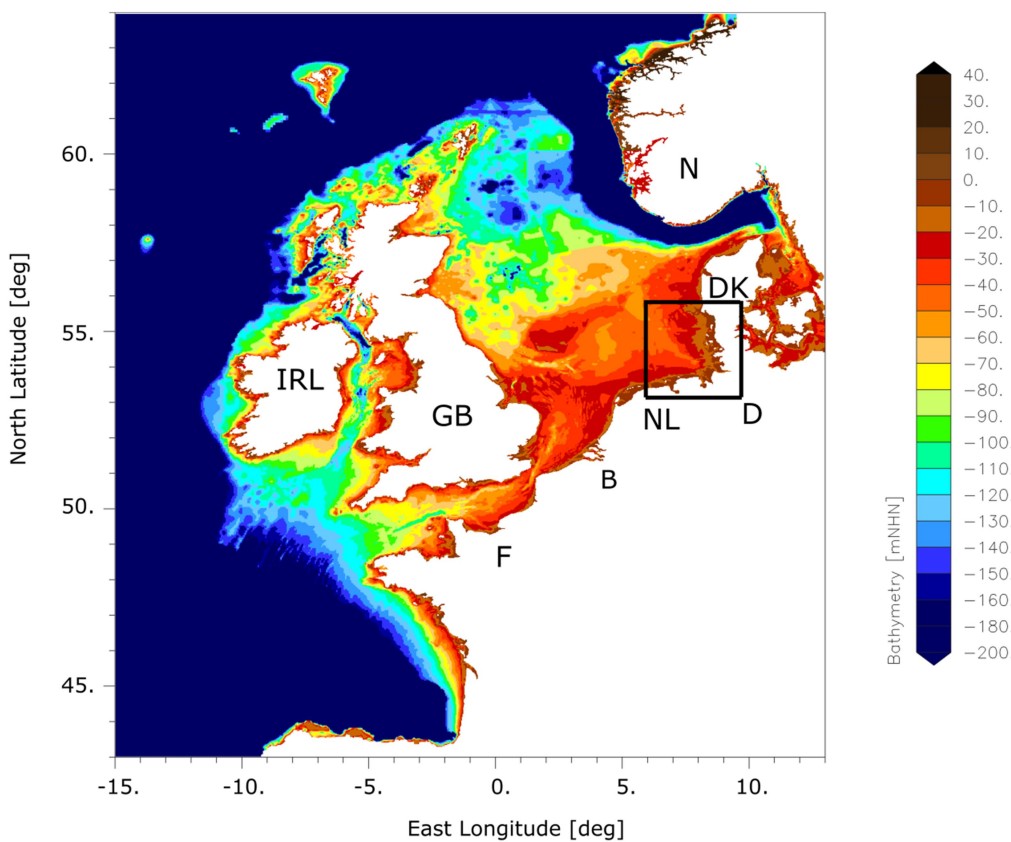

**Figure 1: Model domain of the DCSMv6FM model. The black box marks the German Bight. The unit mNHN denotes metres above the German datum which is a good approximation of mean sea level.**

The model is based on the hydrodynamic numerical model UnTRIM² (Casulli, 2008), which solves the three-dimensional shallow water equations and the three-dimensional transport equation for salt, suspended sediment and heat on an orthogonal unstructured grid (Casulli and Walters, 2000). In the model set-up used here the transport of suspended sediment is not calculated as it is computationally intensive and not of primary relevance to tidal dynamics. To account for baroclinic processes the simulations are carried out in 3D. An advantage of the UnTRIM²-method compared to its predecessor UnTRIM is the subgrid option. This option allows to describe the bathymetry at a higher resolution compared to the computational grid (Sehili et al., 2014). The algorithm, which was derived by Casulli (2008) and Casulli and Stelling (2011), represents correctly the precise mass balance in regions where wetting and drying occur. The computational grids are permitted to be wet, partially wet or dry. This implies that no drying threshold is needed (Sehili et al., 2014).

The computational grid has a resolution of 5 km at the open boundary, 300 m in the coastal areas and 100 m in the estuaries. The subgrid technology is used in the estuaries and the coastal zone with a resolution of 40 m in the finest parts. Due to the high resolution of the intertidal zone, flooding and drying can be reproduced well in the model (Sehili et al., 2014).

At the open seaward boundary, water level is derived from DCSMv6FM. In this way shallow water effects generated on the shelf are included in the boundary values. Salinity at the open boundary is provided by results of a North Sea model used in the project AufMod (Milbradt et al., 2015). The aim of the project AufMod was to develop a model-based tool to analyse long-term sediment transport and morphological processes. During this project a numerical model of the North Sea was developed. The salinity boundary condition employed is a result of a simulation carried out in this project.

At the upstream boundaries of the estuaries, measured river discharge and a constant salinity is applied. The measured river discharge is provided by the Water and Shipping Authorities, the Hamburg Port Authority and the NLWKN (Hamburg Port Authority; Niedersächsischer Landesbetrieb für Wasserwirtschaft, Küsten- und Naturschutz, 2013) with a temporal resolution of 1 day.

The initial data for salinity in the estuaries are provided by the project KLIWAS (Seiffert et al., 2014). They range from 0.4 PSU near the upstream boundaries to 33 PSU in the mouths of the estuaries. In the outer German Bight a value of 33 PSU is assumed. For the initial conditions of the water level, a constant value of 0 mNHN is set. A sufficiently long initialisation time ensures that the model reaches a dynamical equilibrium before the simulations start. Atmospheric forcing (wind at 10 m and atmospheric surface pressure) is included using of the same reanalysis data as for the shelf model DCSM6vFM (. The bathymetric data used in the German Bight Model are mainly based on data provided by the DHI (Danish Hydrological Institute) and the BSH (Federal Maritime and Hydrographic Agency Germany). Near the coast bathymetric data is updated with results provided by the project AufMod (Milbradt et al., 2015).

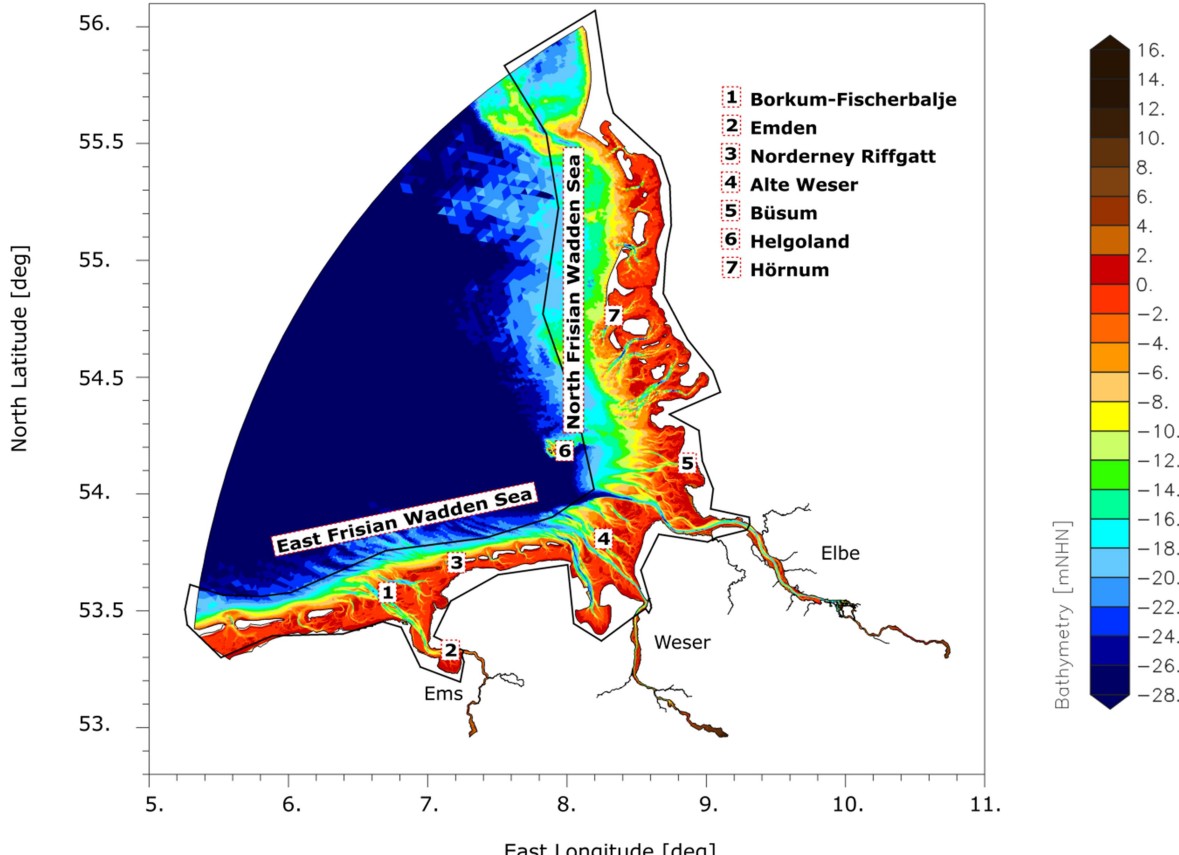

**Figure 2: Model domain of the German Bight Model. The area within the black polygon is used for spatial averaging as described in section 3.4.**

## 2.3 Model validation

In Table 1 the bias (mean deviation between measurement and model) and the root mean square error of the two models compared to measurements for tidal high water, tidal low water, mean tide level and the M2 amplitude are listed for 5 different stations in the German Bight. The stations are marked in Figure 2. The validation period for tidal high water, tidal low water and tidal mean water is a spring-neap-cycle in July 2010 (06.07.2010 – 21.07.2010). For the validation of the M2 amplitude three months are used (03.06.2010 – 01.09.2010). A period of three months comprises the cycle of most relevant tidal constituents in the North Sea. The comparison shows that both models are able to represent basic characteristics of the tidal dynamics. In general, tidal high water is simulated to higher accuracy than tidal low water. The comparable shape of the water level curves between the two models, and measurements can be seen by the example in Figure 3 for the station "Borkum Fischerbalje". Figure 4 shows a "Target Diagram" in which the water levels of the DCSMv6FM and the German Bight Model are compared with minutely measurements of water level at seven stations in the German Bight. The "Target-Diagram" relates the uRMSD* (unbiased Root-Mean-Square-Difference normalized by the standard deviation) and the bias*

(mean deviation between measurement and model normalized by the standard deviation) (Jolliff et al., 2009). The closer the individual points are positioned to the centre, the higher is the model's accuracy. The modelled water levels at the displayed stations are for both models almost all within the inner circle within a range of -0.25 to 0.25 which resembles a RMSE* (Root-Mean-Square-Error normalized with the standard deviation) of 0.25. The only point with a larger RMSE*is the station Emden simulated by DCSMv6FM. Since Emden is located in the inner estuary of the Ems (Figure 2), the water levels are difficult to compute with the shelf model that has a relatively coarse resolution.

**Table 1: Bias and root mean square error for four tidal parameters (tidal high water, tidal low water, mean tide level and M2 amplitude) given in metres at different stations in the German Bight for the German Bight Model and the DCSMv6FM (stations marked in Figure 2).**

| | | thw | | tlw | | mtl | | M2 |
|---|---|---|---|---|---|---|---|---|
| | | Bias | RMSE | Bias | RMSE | Bias | RMSE | Bias |
| Borkum Fischerbalje | GBM | -0.02 | 0.05 | 0.23 | 0.24 | 0.06 | 0.07 | -0.07 |
| | DCSMv6FM | -0.02 | 0.06 | 0.18 | 0.19 | 0.05 | 0.07 | -0.07 |
| Norderney Riffgat | GBM | -0.04 | 0.06 | 0.32 | 0.32 | 0.10 | 0.11 | -0.09 |
| | DCSMv6FM | -0.05 | 0.07 | 0.23 | 0.23 | 0.09 | 0.09 | -0.04 |
| Alte Weser | GBM | -0.10 | 0.12 | 0.28 | 0.28 | 0.08 | 0.09 | -0.09 |
| | DCSMv6FM | -0.07 | 0.09 | 0.22 | 0.23 | 0.06 | 0.08 | -0.04 |
| Helgoland | GBM | -0.13 | 0.14 | 0.31 | 0.31 | 0.07 | 0.08 | -0.12 |
| | DCSMv6FM | -0.22 | 0.11 | 0.25 | 0.22 | 0.06 | 0.08 | -0.07 |
| Hörnum | GBM | -0.12 | 0.12 | 0.20 | 0.20 | 0.06 | 0.07 | -0.11 |
| | DCSMv6FM | 0.10 | 0.05 | 0.00 | 0.07 | -0.01 | 0.04 | 0.01 |

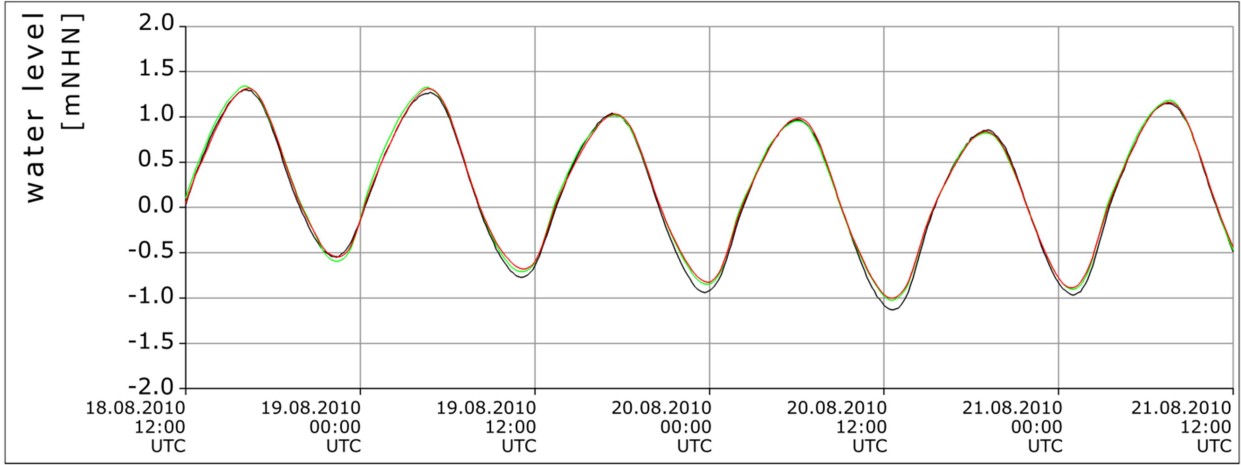

**Figure 3: Water level relative to mNHN (metres above the German datum) at the station "Borkum Fischerbalje" (see Figure 2): Black: Measured data, red: Simulated data with the German Bight Model, green: Simulated data with the DCSMv6FM.**

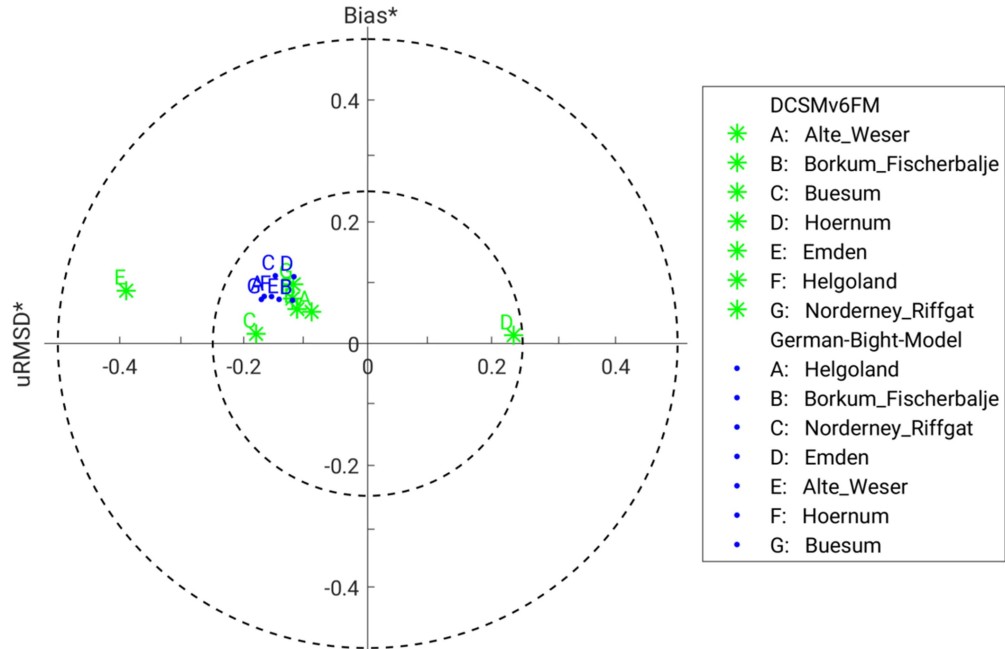

**Figure 4:** "Target Diagram" for the comparison of the DCSMv6FM and the German Bight Model with measured water levels at seven stations along the German Bight (stations marked in Figure 2). The numbers given are normalised by the standard deviation.

## 2.4 Numerical simulations

To investigate the impact of mean sea level rise on tidal dynamics in the North Sea we perform simulations with and without mean sea level rise using the shelf model DCSM6vFM and the German Bight Model (Table 2). For the simulations a period of 3 months (June, July and August 2010) is modelled. The summer period ensures that the results are not influenced by
10   storm surges or extraordinary high river discharge. Note, since wind speeds are generally small in the summer period tide-only simulations would give similar results. Two different mean sea level rises are simulated: 0.8 m and 10 m. The value of 0.8 m lies within the projected range of global mean sea level rise in 2100 of the scenario RCP8.5 reported in the 5th IPCC assessment report (Stocker et al., 2013). To gain a better understanding of the system's response to high water levels we use additionally the mean sea level rise of 10 m.

15   The mean sea level rises are added as constant values at the open boundary of the shelf model DCSM6vFM. We assume that the tides will not change at the open boundary of the DCSMv6FM due to MSLR. This appears to be a reasonable assumption since the study is designed as a conceptual one to investigate the interaction between sea level rise and the representation of the bathymetry in the coastal zone, rather than fully characterising the future development of the tides, which may be altered

by rising sea levels (e.g. Harker et al., 2019). The German Bight Model is forced by water level time series extracted from DCSMv6FM that already include the effects of MSLR on the shelf. Sufficiently long initialisation times in both models ensure that the model reaches a dynamical equilibrium.

In addition to the above mentioned simulations we examine two case studies using the German Bight Model. With the help of these case studies we investigate the effects of resolution-dependent simplifications of shelf models. In case study 1 the estuaries of the rivers Elbe, Weser and Ems are removed from the German Bight Model at the locations where the shelf model DCSMv6FM ends. In these runs (GBM_ref_NE and GBM_80_NE), the main difference to the reference runs is the varied length of the estuaries that means that the volumes of the tidal basins are changed. In addition, no river discharge is included. A simulation of the original reference model (GBM_ref) but with no river discharge is denoted GBM_ref_noQ (Table 2).

In case study 2 the coarser bathymetry of the shelf model DCSMv6FM is mapped onto the model grid of the German Bight Model without estuaries. This simulation is compared to the simulations from case study 1. In this way the only difference is the resolution of the bathymetry. The model still has a high resolution grid, but with a coarse bathymetry mapped onto it as shown in Figure 5. The topography of the coarse bathymetry contains artificial shoals and barriers even in deep channels like the mouth of the Elbe estuary. In some areas the water depth is underestimated and in other parts overestimated (Figure 5).

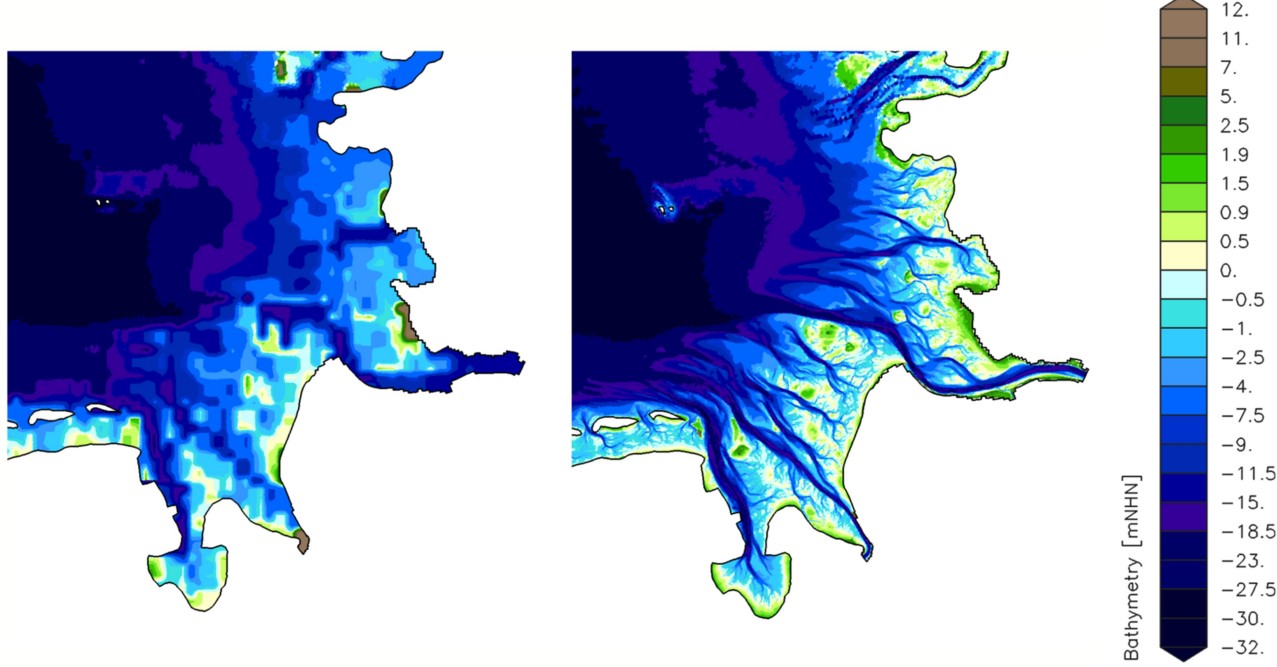

**Figure 5: Bathymetry in the German Bight, left: coarse model bathymetry on the high resolution grid, right: the original highly resolved bathymetry of the GBM.**

**Table 2: Overview of model simulations undertaken.**

| Name | Model setup | MSLR |
|---|---|---|
| **Shelf_ref** | DCSMv6FM | - |
| **Shelf_80** | DCSMv6FM | 0.8 m |
| **Shelf_1000** | DCSMv6FM | 10 m |
| **GBM_ref** | German Bight Model | - |
| **GBM_80** | German Bight Model | 0.8 m |
| **GBM_1000** | German Bight Model | 10 m |
| **GBM_ref_noQ** | German Bight Model, no river discharge | - |
| **GBM_ref_NE** | German Bight Model, no estuaries | - |
| **GBM_80_NE** | German Bight Model, no estuaries | 0.8 m |
| **GBM_ref_NE_CB** | German Bight Model, no estuaries, coarse bathymetry | - |
| **GBM_80_NE_CB** | German Bight Model, no estuaries, coarse bathymetry | 0.8 m |

## 2.5 Analysis of model simulations

The analyses of the numerical model simulations shown in this paper concentrate on the M2 amplitude, mean current velocities and variations in wet area and dissipation rate. The amplitude of the largest tidal constituent M2 (lunar semi-diurnal tide) in the North Sea is estimated by a harmonic analysis of tides (Pansch, 1988), which is based on a Fourier decomposition of the water level time series into harmonic functions of prescribed tidal constituents. The harmonic analysis of tides is applied over the simulation period (3 June – 1 September). The results for wet areas, dissipation rate and mean

current velocities are averages over a full spring-neap cycle (6 July – 21 July). To evaluate wet areas in the model simulations we analyse the mean flooded area at tidal high water. The estimation of the dissipation rate is based on the assumption that a loss of barotropic energy (sum of kinetic and potential energy) is mainly caused by barotropic dissipation. We estimate the dissipation rate $\epsilon$ by computing the divergence of the depth-integrated barotropic energy flux $\nabla_H \overline{F_0}$ where

$$\overline{F_0} = \frac{1}{2}\rho H U^3 + \rho g \eta H U$$

and $\rho$ denotes density, $H$ the total water depth, $U$ the 2-dimensional depth-averaged velocity vector, $g$ the gravitational

acceleration and $\eta$ the deviation from the mean water level. The overbar refers to the depth averaging and the suffix H indicates the horizontal component of the operator. The first term on the right hand side represents the advection of kinetic energy. The second term estimates the barotropic pressure work. For a comprehensive derivation and description of tidal energetics see Kang (2011).

## 3 Model results

### 3.1 M2 amplitude

Figure 6 shows the M2 amplitude and its changes in response to MSLR in the region of the German Bight for both numerical
models. The shelf model DCSMv6FM resembles the results from Pickering et al. (2012). It shows an increase of the M2
amplitude in the German Bight for the two MSLR scenarios. The German Bight Model shows a different response for the
MSLR of 0.8 m. The amplitude increases only in small areas within the Wadden Sea and decreases offshore of the North
Frisian Wadden Sea. This behaviour is comparable to the results of Ward et al. (2012). For the scenario with a MSLR of
10 m the German Bight Model shows like the shelf model an increase of the M2 amplitude in the entire German Bight. A
closer comparison of the responses of both models to the MSLR of 10 m reveals that the differences (compare Figure 6f and
Figure 6e, see supplementary Figure S1b) in the region offshore of the North Frisian Wadden Sea are smaller, but in a
comparable order of magnitude as in the case of MSLR 0.8 m (compare Figure 6d and Figure 6c, see supplementary Figure
S1a). For completeness, supplementary Figure 2 shows the phase lag of the M2 in the German Bight Model and in the
DCSMv6FM in the reference and the according changes due to sea level rise of 0.8 m and 10 m. The changes in the phase
lag of M2 indicate that the tidal wave propagates faster in the simulations with MSLR due to the increased water depths. To
explore the reasons for the observed differences between the shelf model and the German Bight Model two case studies are
conducted and investigated for a MSLR of 0.8 m.

### 3.2 Case Study 1: Removing the estuaries

Since the estuaries are not included in the shelf model the volume of the tidal basins Elbe, Weser and Ems is different in the
two models.  To study the effect of this difference the estuaries are removed from the German Bight Model. They are cut at
the positions where the DCSMv6FM ends in the estuaries.
Figure 7a gives the results of the changes in M2 amplitude due to the removal of the estuaries in the German Bight Model. In
this figure no mean sea level rise is considered and there is no river discharge in the two runs. The M2 amplitude shows
differences only in the mouth of the Elbe due to the removal of the estuaries. The removal leads to an increase of the M2
amplitude. The response of the German Bight Model without estuaries to MSLR of 0.8 m is displayed in Figure 7b. The
comparison to Figure 6d (GBM with estuaries and MSLR of 0.8 m) shows that only in the outer estuary of the Weser some
differences can be spotted and the general pattern of the changes in M2 amplitude stays the same. Thus the different volume
of the tidal basins due to the missing estuaries in the shelf model DCSMv6FM is not the main reason for the differences of
the two models for MSLR of 0.8 m seen in Figure 6.

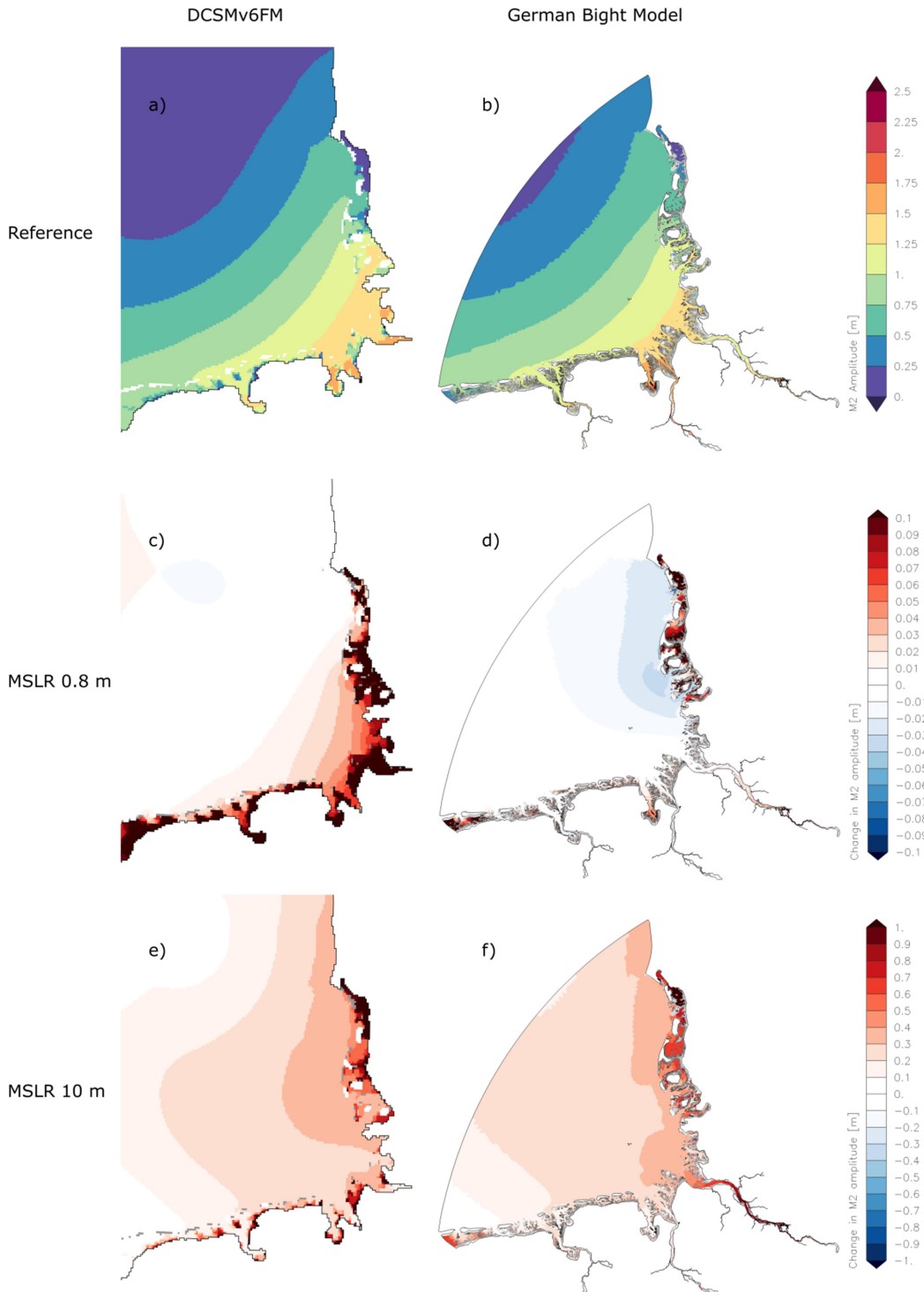

**Figure 6: M2 amplitude and response of M2 amplitude to MSLR (increase in red, decrease in blue, dry areas in grey), left: DCSMv6FM (a, c, e), right: German Bight Model (b, d, f); first row: Reference; second row: MSLR of 0.8 m; third row; MSLR of**

**10 m; a) Shelf_ref, b) GBM_ref, c) Shelf_80 - Shelf_ref, d) GBM_80 - GBM_ref, e) Shelf_1000 - Shelf_ref, f) GBM_1000 - GBM_ref.**

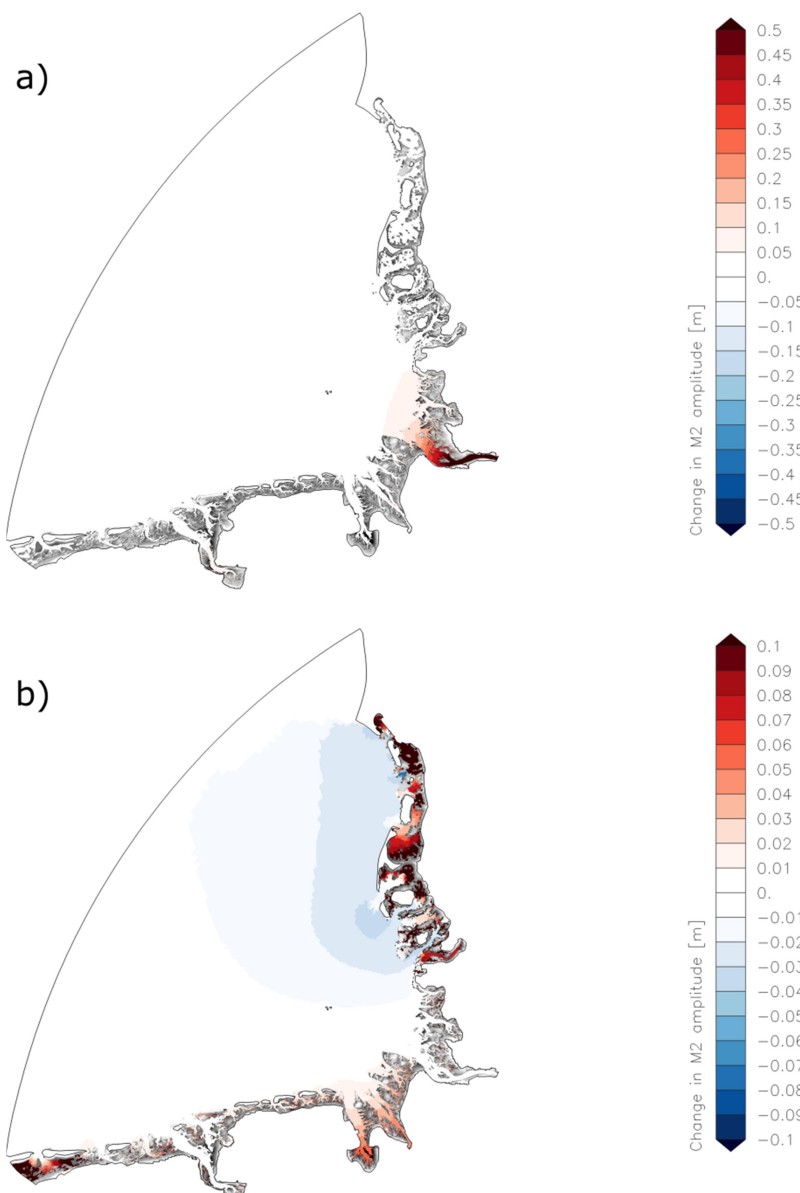

5  **Figure 7: a) Changes in M2 amplitude due to the removed estuaries (increase in red, decrease in blue, dry areas in grey) (GBM_ref_NE – GBM_ref_noQ). b) Changes in M2 amplitude due to MSLR 0.8 m in the German Bight Model without the estuaries (increase in red, decrease in blue) (GBM_80_NE - GBM_ref_NE).**

### 3.3 Case Study 2: coarse shelf model bathymetry

Due to the limited resolution of the shelf model the complex bathymetry in the coastal zone cannot be represented in detail. In this second case study the effect of a coarse bathymetry is investigated by interpolating the coarser shelf model bathymetry onto the high resolution model grid of the German Bight Model.

5   Figure 8a shows the changes in M2 amplitude due to the coarser resolution of the adopted shelf model bathymetry. As a result of the altered bathymetry the M2 amplitude decreases in the inner German Bight. The largest decrease can be detected in the mouth of the Elbe estuary. In contrast to case study 1 the changes are not restricted locally.

The response to MSLR of 0.8 m is shown in Figure 8b. The increase of M2 amplitude in the German Bight in this case study is now comparable to the shelf model response (Figure 6c). Therefore, most of the changes in the shelf model induced by the

10  MSLR of 0.8 m must be due to the coarse bathymetry.

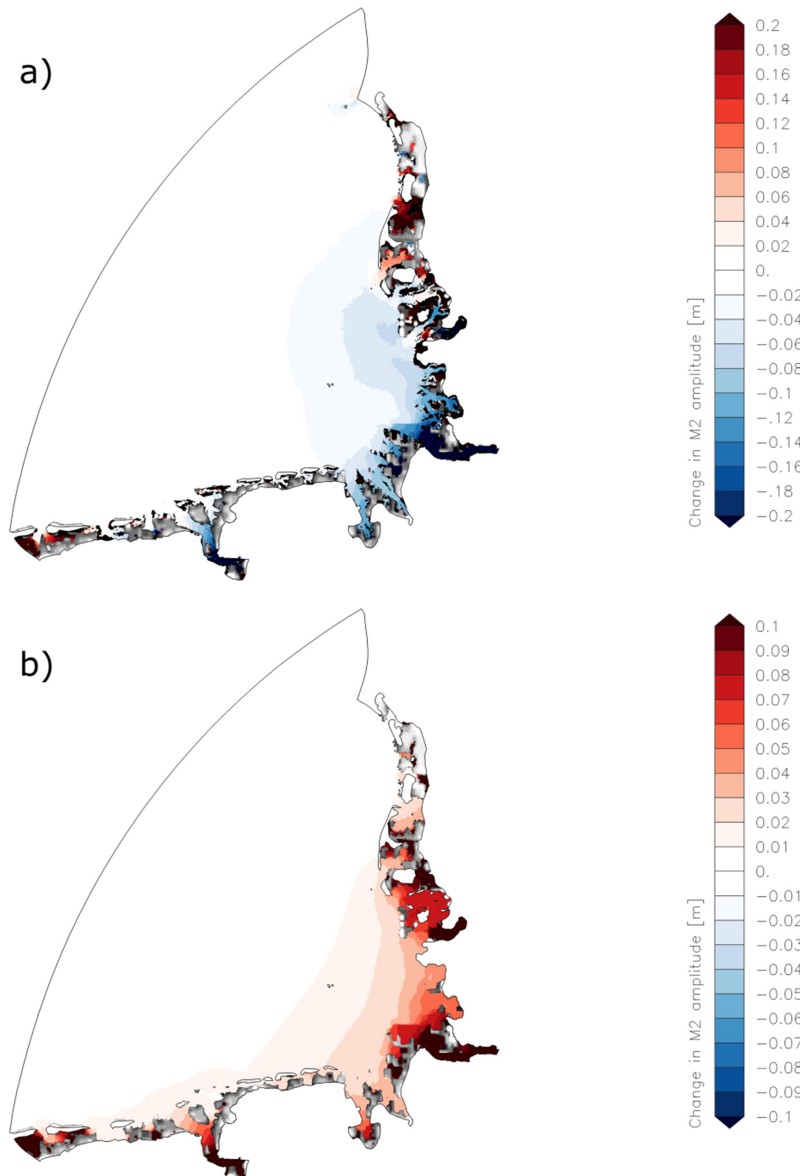

**Figure 8: a) Changes in M2 amplitude due to the coarser bathymetry (increase in red, decrease in blue, dry areas in grey) (GBM_ref_NE_CB – GBM_ref_NE). b) Changes in M2 amplitude due to MSLR 0.8 m in the German Bight Model with coarse bathymetry (increase in red, decrease in blue) (GBM_80_NE_CB – GBM_ref_NE_CB).**

## 3.4 Wet areas, dissipation rate and current velocities

To further investigate the reasons for the similarity of the response to 0.8 m MSLR of case study 2 compared to the shelf model DCSMv6FM (Figure 6c and Figure 8b) we analyse the mean flooded area at tidal high water (wet areas) and the

dissipation rate in different runs (Table 3). Since we expect the main differences to be in the shallow part of the German Bight, we determine the average values within the area of the Wadden Sea including shallow parts out as far as 20 m depth (shown by the polygon in Figure 2).

The numbers in Table 3 show that wet areas increase due to MSLR. In both situations with the highly resolved bathymetry and in the coarsely resolved bathymetry, the gain of wet areas due to MSLR of 0.8 m is about the same. In contrast, the change in dissipation rate due to mean sea level rise differs between the runs. With the fine bathymetry the dissipation rate increases by about 21 % ($0.6 \times 10^{-3}$ W/m$^2$) whereas with the coarse bathymetry it increases only by about 7% ($0.2 \times 10^{-3}$ W/m$^2$).

**Table 3: Mean flooded area at tidal high water (wet area) in the shallow part of the German Bight out to 20 m depth and dissipation rate averaged over that region (see Figure 2).**

|  | wet area [$10^9$ m$^2$] | dissipation rate [$10^{-3}$ W/m$^2$] |
|---|---|---|
| **GBM_ref_NE** | 15.90 | 2.9 |
| **GBM_80_NE - GBM_ref_NE** | 0.22 | 0.6 |
| **GBM_ref_NE_CB** | 15.91 | 3.0 |
| **GBM_80_NE_CB - GBM_ref_NE_CB** | 0.24 | 0.2 |

Figure 9a and c show the mean current speed (depth averaged and analysed over a spring-neap-cycle in July 2010) in the reference case and the change of mean current speed due to MSLR of 0.8 m in the fine bathymetry with removed estuaries. Figure 9b and d shows the same for the coarse bathymetry. In the fine bathymetry the mean current speed increases due to mean sea level rise in coastal areas, especially in the tidal channels, almost everywhere in the near-shore parts of the German Bight. In the coarse bathymetry the change of mean current speed has a different pattern. In general the course of the channels is less distinctively represented and increases in mean current speed are not as pronounced as in the fine bathymetry. These results are consistent with the smaller increase of dissipation rate in the coarse bathymetry compared to the case of the fine bathymetry, since dissipation rate strongly depends on speed.

The significance of the shallow areas near the coast for the dissipation of energy is illustrated in Table 4. Besides the dissipation rate averaged over the shallow part of the German Bight out to 20 m depth (area within the black polygon in Figure 2) the table contains also dissipation rates averaged over the entire model domain of the German Bight Model excluding the estuaries. In general, the domain-averaged dissipation rates are smaller than dissipation rates averaged over the shallow parts. In the reference simulation and the run with 0.8 m MSLR the dissipation rate in the shallow part is higher by a factor of approximatively 1.8.

Furthermore Table 4 includes also wet areas and dissipation rates of the simulation with 10 m MSLR. The increase of wet area from 0.8 m MSLR to 10 m MSLR is less than the increase of wet area from the reference run (no MSLR) to the run with 0.8 MSLR. Dissipation rate in the model run with 10 m MSLR decreases in comparison to MSLR of 0.8 m. In the

shallow part of the German Bight out to 20 m depth it also decreases in comparison to the reference simulation. Generally, mean currents decrease in the channels in the model run with 10 m (Figure 10). This is consistent with the smaller values of dissipation rate.

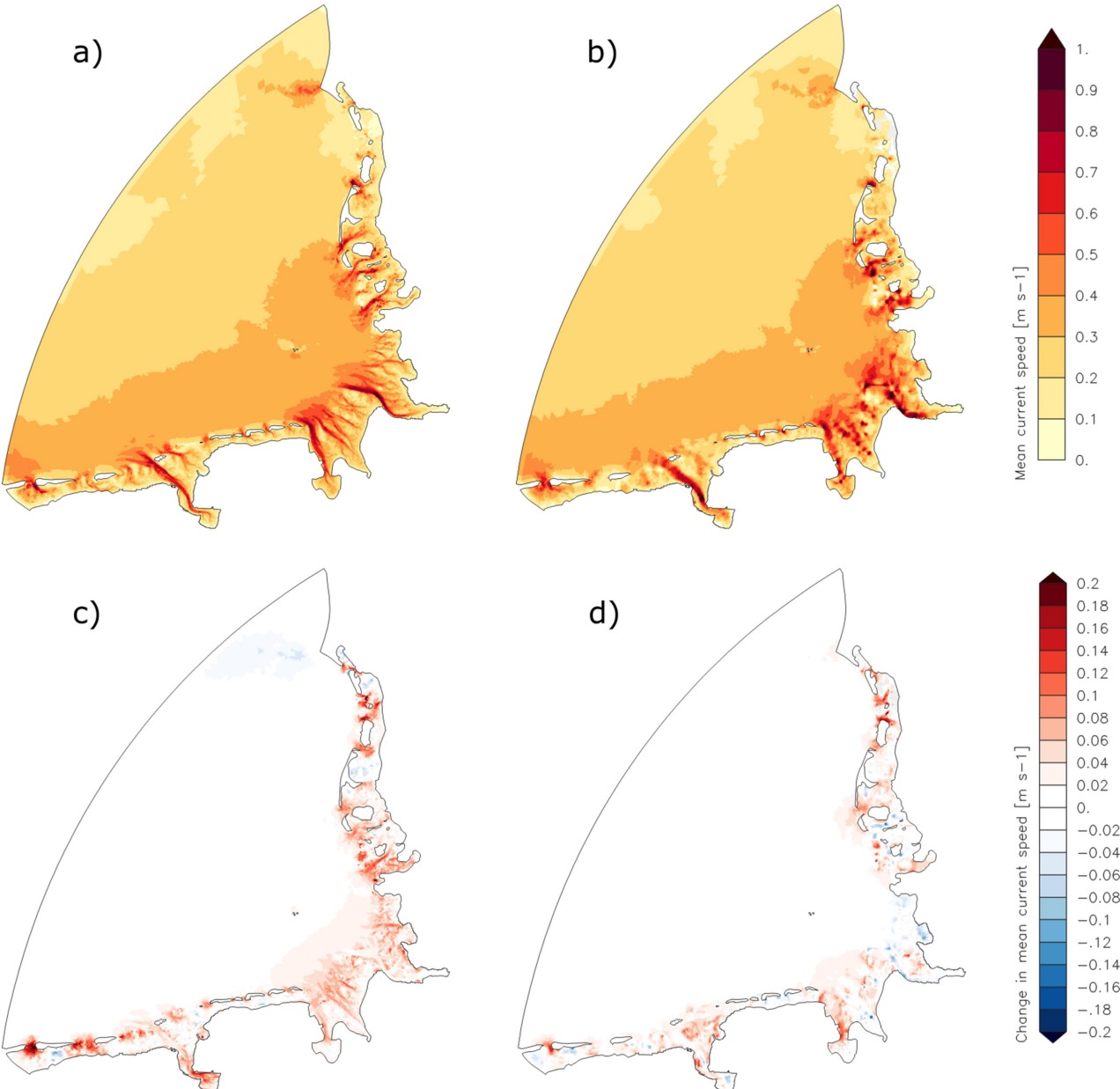

5    **Figure 9:  a) Total mean current speed (depth averaged and analysed over a spring-neap-cycle in July 2010) without MSLR in the high resolution bathymetry (GBM_ref_NE) and b) in the coarser bathymetry (GBM_ref_NE_CB); c) Change in total mean current speed in the high resolution bathymetry (GBM_80_NE - GBM_ref_NE) and d) in the coarser bathymetry due to MSLR of 0.8 m (GBM_80_NE_CB - GBM_ref_NE_CB).**

**Table 4: Mean flooded area at tidal high water (wet area) in the shallow part of the German Bight out to 20 m depth, dissipation rate ϵ_S averaged in the shallow part of the German Bight out to 20 m depth and dissipation rate ϵ_G averaged over the entire German Bight Model domain excluding the estuaries.**

| | wet area $[10^9\,m^2]$ | dissipation rate $[10^{-3}\,W/m^2]$ | | |
| --- | --- | --- | --- | --- |
| | shallow part | shallow part $\epsilon_S$ | domain $\epsilon_D$ | $\epsilon_S/\epsilon_D$ |
| **GBM_ref** | 15.88 | 2.8 | 1.6 | 1.75 |
| **GBM_80** | 16.11 | 3.4 | 1.9 | 1.79 |
| **GBM_1000** | 16.21 | 2.5 | 1.8 | 1.39 |

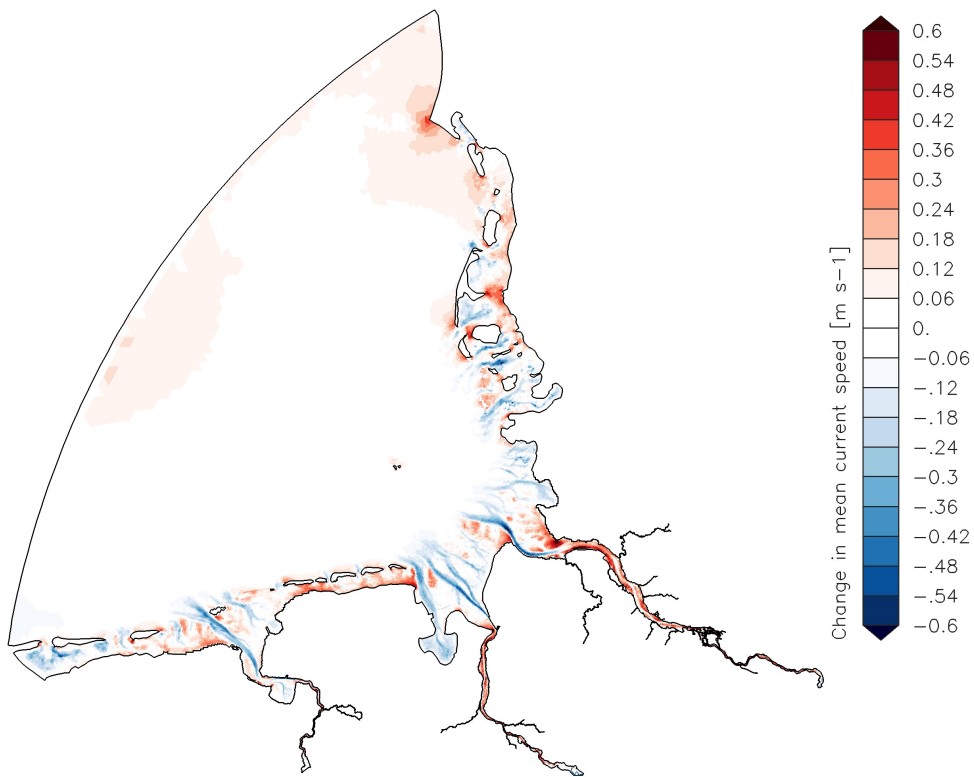

**Figure 10: Change in total mean current speed (depth averaged and analysed over a spring-neap-cycle in July 2010) due to MSLR of 10 m (increase in red, decrease in blue) (GBM_1000-GBM_ref).**

**4 Discussion**

In this study we compare the response of two different kinds of models to mean sea level rises of 0.8 m and 10 m in the German Bight. The coarser shelf model DCSMv6FM and the finer German Bight Model respond in different ways to mean sea level rise (MSLR). To identify the reasons for the different responses we adjust the German Bight Model in two case studies in order to more closely resemble the shelf model and repeat the simulations with MSLR of 0.8 m. In the first study the estuaries are excluded from the model domain. While the reduced volume of the tidal basins due to the shortened estuaries explains the locally increased M2 amplitude, it does not explain the different responses of the two models seen on a larger scale. In the second study the coarse bathymetry of the shelf model is mapped onto the fine model grid of the German Bight model. With this second study, the German Bight Model was found to respond in a similar way to as the shelf model DCSMv6FM. Thus it is mainly the different resolution of the bathymetry used in the two models, which leads to the different responses.

Pelling et al. (2013) explained the different response to MSLR of two shelf models by means of different dissipation behaviour due to newly flooded cells outside the former model boundary in one of the two models. The boundaries of the Shelf Model DCSMv6FM and of the German Bight Model are defined in a way that dikes cannot be overflowed, i.e. no new cells can be flooded behind dikes in the former hinterland like in the study of Ward et al. (2012). However, owing to the drying and flooding algorithm implemented, the DCSMv6FM and the German Bight Model are able to flood new cells in the dike foreland when mean sea level rises. Following the argumentation of Pelling et al. (2013), one explanation for the different response to MSLR in the finer German Bight Model and the coarser shelf model DCSMv6FM could be that less new area is flooded in the shelf model when mean sea level rises and thus less highly dissipative area exists in the shelf model. In this way the larger dissipative areas in the fine model would be an explanation for the weaker increase and in some regions decrease of the M2 amplitude in the fine model. The analysis of wet areas (Table 3) in the different case studies, however, does not support this explanation. The changes in wet area due to a MSLR of 0.8 m in the model with fine bathymetry (case study 1) and the model with coarse bathymetry (case study 2) do not differ significantly. Nevertheless, the change of dissipation rates due to MSLR of 0.8 m is different in the model runs with fine or coarse bathymetry. In the fine bathymetry model dissipation rate averaged over the region of the Wadden Sea including the shallow part out to the 20 m depth increases, whereas it almost stays constant in the coarse bathymetry model. The larger increase in dissipation rate in the fine bathymetry model results mainly from overall increased current speeds. In the coarse bathymetry model this increase in current speeds cannot be seen to the same extent. The coarse bathymetry contains many artificial shoals and barriers. In this case, many channels in the Wadden Sea do not allow a continuous flow of water. This leads to the differences in mean current speed and its response to MSLR in the coarse model.

These results suggest that a sufficiently fine resolution of shallow regions such as the Wadden Sea is needed in hydrodynamic models for the most accurate representation of tidal dynamics and its response to MSLR possible. In this respect, shelf models as used by Ward et al. (2012) and Pickering et al. (2012) are only within limits suited to draw conclusions for the tidal response to MSLR in shallow areas such as the German Bight. One question that needs further research is how fine the bathymetric resolution should be to estimate the response of tidal dynamic to MSLR correctly. A sensitivity study varying the resolution of the computational grid systematically could provide further insight into this open question. The subgrid technology used in the regional German Bight Model, however, already allows to specify bathymetric details at a very high resolution. As shown in Sehili et al. (2014) different resolutions of the computational grid (within a certain range) do not influence the simulated results when using a subgrid. Their conclusion is that a relatively coarse resolved computational grid yields similar results to a finer resolved computational grid when using the same very fine resolved subgrid information. Thus we suppose that a different resolution of the computational grid would not change the basic results. To confirm this supposition further studies on the role of bathymetric subgrid information in combination with MSLR are needed.

The increase in mean current speed at 0.8 m MSLR can be explained by an increased ratio of flood volume to cross-sectional areas of the tidal inlets (Wachler et al., submitted). The MSLR induced change in ratio of flood volume to tidal inlet cross sectional area depends on the geometry of the tidal basin, e.g. on the ratio between the area of intertidal flats and channels. The tidal basins in the Wadden Sea of the German Bight are characterised by larger intertidal flat areas relative to the channel areas (Ferk, 1995; Spiegel, 1997). Due to these geometric characteristics, with rising mean sea level the flood volume increases more than the cross section of the tidal inlet resulting in higher current speeds in the tidal inlet system.

In contrast, in the simulation with 10 m MSLR mean current speed decreases in the channels. We suppose that the decrease of mean current speeds is due to the much higher increase of water levels compared to the scenario with 0.8 m MSLR. The water extends up to the model's boundary and can only accumulate vertically but cannot overflow new areas. Wetting and drying do not take place any longer. Unlike in the case of 0.8 m MSLR the cross-sectional areas of the tidal inlets increase considerably more such that the ratio between flood volume and the tidal inlet cross sectional area decreases.

Several points can be mentioned concerning the extent to which model simulations such as described in this paper can be applied to estimate the tidal response to MSLR in a real future. In this study we add MSLR as a constant value to present day boundary conditions. Thus changes in the response of global tides to MSLR are not included. Harker et al. (2019) find that for the Australian Shelf the differences between considering the change of global tides to MSLR and using present day tides are not negligible. For the German Bight it is not clear how large this difference would be. Further studies are needed.

Another remark relates to the assumption of unchanged bathymetries in the case of mean sea level rise. For example, a vertical growth of tidal flats is expected due to MSLR (Hofstede, 2002, van Maanen et al., 2013). For a study considering changes of the Wadden Sea bathymetry in combination with MSLR see Wachler et al. (submitted). Furthermore, since dikes cannot be overflowed in both numerical models used, especially the simulations with 10 m MSLR especially do not represent how the system would react in the real world, in which dikes are usually not high enough to retain such high water

levels. These simulations are included here only to gain a better understanding of the system response to increased water levels caused by mean sea level rise.

## 5 Conclusions

In flat coastal areas such as the Wadden Sea in the German Bight, the small-scale representation of the bathymetry plays a crucial role in the estimation of changing tidal dynamics in response to mean sea level rise. The dissipation rate in the region of the Wadden Sea is considerably higher than in deeper areas. Thus these shallow areas must be sufficiently resolved. Depending on the research question and the geographic area of interest, it is important to select the model setup in such a way that all relevant processes are sufficiently taken into account. For investigating the response of the wider North Sea, the use of a shelf model with lower resolution might be sufficient. However, to draw conclusions for coastal stations it is necessary to use numerical models that resolve coastal bathymetry and the shoreline (e.g. including estuaries) as well as possible.

## 6 Data availability

Data and results in this article resulting from numerical simulations are available upon request from the corresponding author.

## 7 Author contribution

Caroline Rasquin worked on the simulations, analyses and figures. Rita Seiffert and Benno Wachler participated in the analyses and the interpretation of the results. Caroline Rasquin and Rita Seiffert wrote the manuscript with contributions from Benno Wachler. Norbert Winkel was involved as a scientific expert, supervised the study and gave input for writing and revision of the paper.

## 8 Competing interests

The authors declare that they have no conflict of interest.

## 9 Acknowledgements

This work has been carried out within the framework of the Network of Experts financed by the German Federal Ministry of Transport and Digital Infrastructure. We thank all our co-workers at the Federal Waterways Engineering and Research Institute in Hamburg for their continuous support. Special thanks go to Günther Lang, Frank Kösters, Julia Benndorf and

Elisabeth Rudolph for inspiring discussions. We also thank three anonymous reviewers and the editor Philip Woodworth for their careful reading and many valuable comments to improve the manuscript.

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
