# Peer review of "The significance of coastal bathymetry representation for modelling the tidal response to mean sea level rise in the German Bight"

_Ocean Science, 2019_

## Referee Comment (RC1) · Anonymous Referee #1 · 24 Jul 2019

**1 General comments**

In this work, the authors describe the impact of different bathymetric resolution on the tidal response to sea level changes in the German Bight. The authors use two different model setups to compare the responses to sea level changes and investigate the differences in response between the two models with different sensitivity experiments. The paper is well written and the figures presented are of high quality. Parts of the manuscript could benefit from some improved reasoning. The methodology could be expanded and clarified by including a more detailed description of the tide relevant methods in the models. The analysis appears to be carried out thoroughly, though

some further model – tide gauge evaluation should be added. I am not quite sure that the authors reach the correct conclusions with regards to the large sea level change scenario, however, apart from this point, their conclusions are sound.

**2  Specific comments**

P1 Line 23: The first sentence feels clumsy – consider omitting (you explain in more detail later on anyway) or at least add a reference for this (quite bold) statement

P1 Lines 23 – 28:  what is the point you are trying to make by mentioning the acceleration in MSLR? Maybe it would be more helpful to look at total increases in SLR? This is what really will affect coastal areas. Are there more recent references?

P1 Line 28-29: worth including a reference and eluding to the reasons for this.

P1 Line 28:  highlight why changes in the tides are important – water level variations, extreme water levels, species distributions, changing currents, etc

P1 Line 29 – P2 Line 7:  are the changes you mention here due to changing tides or changing sea level? If the latter then they feel slightly out of context.

P2 Line 25:  what sort of shelf models are you referring to here?  Tide models? OGCMs?

P3 Shelf Model description(s):  Given that your work is all about tides it would be helpful to include more information on the calculations of the tides in the model.

How is bed roughness dealt with? Internal tides? Is this a 2D or 3D model? What happens at the open boundaries?

P3 Line 19: are you really using GOT00.2? The latest version is GOT4.7

P4 Line 5: why would you want sediments in your model?

P4 Line 7: What parameters are used in the subgridscale option? How does this feed into the shallow water equations? Do both models have this option? How does turning this option on and off affect the results?

Table 1: It would helpful to include a root mean square error here as this gives an additional measure of the absolute model error rather than the relative error you get with the bias.

P6 Line 7: explain RMSE*?

P6: Given that you mainly discuss changes in M2 amplitude later on, it would be helpful to include an evaluation of the model performance in simulating the tides against the amplitudes at the stations and possibly also against a product such as TPXO or FES. This is probably more important than including an analysis of the water levels as it is not clear through which processes the water level errors arise (i.e. your model could perform very well at simulating tides but not for storms or vice versa).

P7 line 12 – 13: This statement is not clear to me – why do you add SLR at the open boundary rather than over the whole domain? Is this the case for both models?

Can you explain in more detail how the model deals with flooding areas, i.e. the
wetting and drying scheme? How does that work at low water and sea level rise?

Surely, with SLR more normally dry areas are flooded and more energy is lost there?

P7 Simulations: How do you deal with open boundary tidal forcing with sea level rise? Harker et al. (2019) show that using present day tides at the model open boundaries leads to large changes in tidal responses on the Australian Shelf – can you comment on this for your work? Similarly, global studies such as Mawdsley et al. (2015) show global changes in water levels. Work by Wilmes et al. (2017) show global tidal responses to large-scale sea level changes on the levels of your 10 m simulation.

Do you allow for flooding in the large SLR scenario? What coastal defences do you assume?

Figure 4: Why does Emden have such a large error?

Figure 3: what are the units? Explain mNHN in the caption

P10 Line 3: How do you calculate tidal amplitudes?

P10 line 4-5: omit sentence

Figure 6. Which model performs better at the present-day tides?

P19 lines 4-5: '"The response to a MSLR of 10 m is more comparable in both models" – the differences between the two model setups are probably on a similar magnitude or larger as for the 0.8m SLR scenario, however, they are masked by the larger-scale differences occurring on the whole shelf area (whereas for a 0.8 m

SLR the shelf-scale changes are much smaller). I would, however, argue that there are pretty big differences between Figure 6 e and f, especially around the islands in the southern part of the domain where the difference amounts to over 1 m in amplitude.

P19 lines 32-33: I'm not sure I agree – see comment above. The differences locally are pretty significant. Your large-scale, regional or shelf-wide responses are similar, but then arguably they are also similar (i.e. not very much happens) for small sea level increase.

P20 line 24: "For higher mean sea level rise scenarios (10 m) the resolution of the bathymetry is less important" see comments above. I would conclude that, if looking at complex coastal areas such as the German Bight, highly resolved near-shore bathymetry is important for assessing the impact of sea level changes in these complex areas as the local responses can differ from the regional, offshore tidal changes. This is the case whether the forcing is large or small.

**3  Technical comments**

P2 Line 21: "1m" – space needed; also check remainder of the document

P2 Line 23: defence -> defences

P2 Line 28: After "Thus" add a comma

P10 Lines 3 – 4: Figure 6 shows for both numerical models the M2 amplitude and its changes in response to mean sea level rise in the region of the German Bight.

-> Figure 6 shows the M2 amplitude and its changes in response to mean sea level rise in the region of the German Bight for both numerical models.

Table 2: You have quite a lot of different cases – remembering what case study 1 referred to later on in the manuscript is hard – why not label them something intuitive like GBM_ref_CS1 -> GBM_ref_NE and GBM_80_CS2 -> GBM_80_NE_CB

Figures 7 onwards: rather than having lots of single plot figures it would be better to condense your individual images into one or two figures with more subplots like you have done in Figure 6.

P13 lines 8 - 9: Similar to the shelf model the M2 amplitude increases in this case study in the German Bight. -> The M2 amplitude increases in the German Bight in this case study are now similar to the shelf model changes.

---

## Referee Comment (RC2) · Anonymous Referee #2 · 8 Aug 2019

Suitability: The subject of the paper, i.e. the study of the significance of coastal bathymetry representation for modelling the tidal response to mean sea level rise in the German Bight falls within the fields covered by Ocean Sciences.

Summary: The manuscript uses together a shelf and regional models to investigate the effect of the bathymetric resolution and the estuary morphology on the modelled tidal response to sea level rise in the German Bight. It shows that the tidal response to moderate sea level rise (0.8m) is strongly affected by the bathymetric resolution, while it is much less sensitive for much higher sea level rise (10m).

General comment: The present work, is, to my knowledge the first study quantifying

the effect of the bathymetric resolution on the estimation of tide changes induced by sea level rise. It highlights the potential limits of existing nearshore tide changes projections. To me, the manuscript deserves to be published, but under the condition of major improvements. Indeed, the paper lacks a clear description of the configuration used in the simulation (see my major remarks below). In addition, the model validation could be reinforced in terms of tide changes validation (see my suggestion in below). The regional model has a spatially varying resolution with a minimal grid size of 100 m in the estuaries. As the paper focusses on the effect of the coastal bathymetry representation, then: (1) a sensitivity analysis of the regional model to the bathymetric resolution would strongly reinforce the manuscript and its impact, (2) some recommendations on the relevant grid size to use in the German Bight would be welcome.

Major remarks

1. Methods and model description. The studies relies on two models. In the description of the models (sections 2.1 and 2.2), it seems that the simulations are done taking into account meteorological forcing, offshore salinity and river discharge. No information is provided on these conditions. In addition, if the regional model accounts for the salinity on its offshore boundary, we can guess that the river salinity is also accounted for. Nothing is said, no value is given. Thus, the manuscript really required improvement on the description of the simulations (input conditions) and justification. Indeed, for the purpose of the study, why using 3D baroclinic simulations rather than 2DH simulations? Nothing is said.

2. Validation. First, the manuscript should provide the validation period over which they validate the tide. Second, even if they do not state it, the authors assume indirectly that the regional model (at high resolution) is better than the shelf model (at low resolution) to reproduce tide changes induced by sea-level rise. To support this assumption, a comparison of observed past tide change trends (using for instance literature results on tide gauges located in the study area) and results obtained simulating an additional and more moderate sea level rise scenario of e.g. 0.2 or 0.3 m, would be useful, with all

the limits that such comparison has (additional mechanisms can contribute also modify the tide). But, as in Schindelegger et al. (2018), this would reinforce the paper.

3. Sensitivity of the tide changes to the bathymetric resolution. The manuscript would strongly benefit from a real sensitivity study of the tide changes to the bathymetric resolution, by investigating different bathymetric resolutions with the regional model, and not only the one corresponding to the shelf model. This would allow identifying if there is a bathymetric resolution below which there is no further improvement. Such result would allow the authors to make recommendations for the German Bight, and would strongly increase the impact of the work.

References

Schindelegger M, Green JAM, Wilmes S-B, Haigh ID (2018) Can we model the effect of observed sea level rise on tides? J Geophys Res Oceans. https ://doi.org/10.1029/2018J C0139 59

"On-line" Remarks

- P1-Line 27: "flat" -> in most of the paper, the authors use "flat". I think that "low lying" is more relevant

- P1-Line 27: add a reference to figure 1 and figure 2 (German Bight).

- P1-Line 29: "estuaries." -> reference?

- P1-Line 29 -> P2, line 5: would better fit in the discussion section? Or remove it?

- Figure 4: hard to see the green star and blue points → make a zoom for ∼RMSE=0 to 0.5

- P15, line 5: make clear what is the mean current velocity. Is it the M2 - depth averaged current velocity? Is it the M2 depth averaged current velocity averaged over a given period (and is so, which period?)

- P15-Line 17: "approx.." -> "approximatively".

- Table 4: the readability could be improved by putting on the same line "global" and "epsilon G" (see the 4th column of the table).

- P19-line 30 -> line32: "In this . . .Bight.": this sentence is not clear to me -> reformulate.

- P20, line 1 -> line 7: I suggest to avoid providing results/conclusion based on a reference with the status "in prep". I would advice to reformulate this part as a perspective, and explain that Wachler et al. already start to explore the reasons of increase or decrease of velocity depending on the amount of sea level rise.

- P20, line 19: same suggestion, reformulate to just say that Wachler et al. (in prep) explored the bathymetric changes issue.

Specific Ocean Sciences questions

1. Does the paper address relevant scientific questions within the scope of OS? Yes

2. Does the paper present novel concepts, ideas, tools, or data? Not really, it mainly explores the effect of some modelling issues that we expect could play a role on previous model results.

3. Are substantial conclusions reached? Fair

4. Are the scientific methods and assumptions valid and clearly outlined? Not completely

5. Are the results sufficient to support the interpretations and conclusions? Not completely

6. Is the description of experiments and calculations sufficiently complete and precise to allow their reproduction by fellow scientists (traceability of results)? No

7. Do the authors give proper credit to related work and clearly indicate their own new/original contribution? Yes

8. Does the title clearly reflect the contents of the paper? Yes

9. Does the abstract provide a concise and complete summary? Could be more concise

10. Is the overall presentation well structured and clear? Mainly yes

11. Is the language fluent and precise? Mainly yes

12. Are mathematical formulae, symbols, abbreviations, and units correctly defined and used? Yes

13. Should any parts of the paper (text, formulae, figures, tables) be clarified, reduced, combined, or eliminated? Yes

14. Are the number and quality of references appropriate? Yes

15. Is the amount and quality of supplementary material appropriate? No supplementary material

---

## Referee Comment (RC3) · Anonymous Referee #3 · 20 Sep 2019

From my perspective, the paper addresses a highly relevant issue, namely the conflicting results of models on responses of tidal dynamics to changes in mean sea level described in the literature. The present study provides insight and offers a convincing approach and explanation for these differences. It perfectly fits into the topic of this journal and deserves publication.

In principle, the manuscript is well written and I only have a few technical comments I would ask the authors to consider:

1. In the Introduction the sentence, "The German Bight located in the south-east of the North Sea with its flat coastal areas could be especially vulnerable" appeared a

bit surprisingly. The authors started with a global perspective and I wondered why the German Bight was the area of choice. In addition, why is the German Bight especially vulnerable compared to other flat and low-lying coasts?

2. As the authors motivated their study by differences in the response of global models, can the conclusions from the German Bight be generalized?

3. Model set-up and experiments need some more explanation, in particular

a. Why did the authors choose to include meteorological forcing and not just used a tide-only simulation, in particular as they said they choose the summer to "ensure that the results are not influenced by storm surges or extraordinary high river discharge" (page 7).

b. The salinity boundary condition (page 5) needs some explanation as I expect that readers are not necessarily aware of what was done in the referenced project.

c. A source for the river discharge data and their time resolution should be provided.

4. Figure 6 and following: The gray color bar should be explained in the caption. I was not able to clearly identify gray areas in the Figures.

5. Page 19, line 29: What exactly is "sufficiently fine"? There are probably substantial changes in bathymetry over time as well. What would the authors then consider a "sufficiently fine" resolution?

---

## Editor Comment (EC1) · Philip Woodworth (Editor) · 22 Sep 2019

Separate editorial comments on 'The significance of coastal bathymetry representation for modelling the tidal response to mean sea level rise in the German Bight' by Rasquin et al.

I decided to post these comments now so that the authors can consider them alongside the comments of R1-R3 without further delay. I note that the authors have not responded already to the reviewers' comments.

It is clear from R2 in particular, and maybe R1, that there are both several major and

many minor improvements needed to the paper, although they should be quite feasible to do. So the authors are encouraged to make a new version which, because of the likely changes, I might pass by the reviewers again.

My own general impression is that I found the text not to have been written as well as it should. It is perfectly understandable but it would have been best to have passed it by a native English speaker before submitting the paper. Therefore, I do not understand the comments of a couple of the reviewers that is was 'well written'. I have scribbled many comments on a paper copy of the draft and sent a pdf separately to the lead author (caroline.rasquin@baw.de).

Some particular comments are:

p1, line 8 (and other places) - is the German Bight really in the SE of the North Sea rather than the east.

figures 1 and 2. These have several errors. Fig 1 has degrees longitude and latitude swapped. But anyway they should read North Latitude (deg) and East Longitude (deg).

In figure 1 deep water bathymetry, shown by dark blue, is a negative number whereas in Figure 2 it is a positive number. In Figure 2 it is hard to read the black names.

Explain somewhere that NHN is the German datum which is a good approximation of MSL.

Table 1. replace commas in the numbers with dots which is more normal internationally.

Figure 3. define MEZ in terms of UT

It is not necessary to have 00:00 in the times.

Figure 3. Why is there a slightly different set of 7 stations used for the 2 models (because of the grids presumably). explain better. It is not easy to spot the grouping of blue A,B etc. when printed - I had to blow up the pdf to see that. Anyway Target-Diagram would be better as 'Target Diagram' and things would be clearer if the maximum radius

was 0.6.

p8, 5 and elsewhere - oscillation volume sounds odd to me although I am struggling to think of something else. perhaps have this sentence read: Therefore, the only difference between models concerns the volume of water exchanged through the tidal cycle, which we call the oscillation volume.

Figure 5. why does this have a different colour scale to Figure 2? It covers almost the same area.

section 2.5 header. Please do not be so cryptic. Perhaps Analysis of Model Simulations. section 3 Model results

p10, 26 - this is not surprising as the volume of water in the estuaries is small.

Figure 6 and others. What is the second b/w scale for? Is that because of the wet/dry areas or what? Anyway it is not explained in the captions. It seems to me it could be just left off.

The paper does not discuss changes in phase lags, only amplitudes. Nothing to say about them?

p15 current velocity should be current speed (velocity is a vector)

Figure 11 and 12 (a) remove the white arrow. You can't have a negative speed.

Somewhere I noted R2 pointing to the recent Schindelegger et al. (2018) paper regarding model validation, and I was reminded of the Harker et al. (2019) paper in Ocean Science concerning the important aspect of whether model tides are allowed to change on an open boundary when MSL changes, and that should be made clear in the present paper.

———————————————————

---

## Author Comment (AC1) · 8 Nov 2019

We thank the first reviewer for his/her effort on the review, the careful reading and for the constructive comments and suggestions to improve the paper. Below you find your comments and our response on each point.

P1 Line 23: The first sentence feels clumsy – consider omitting (you explain in more detail later on anyway) or at least add a reference for this (quite bold) statement

*We have omitted this sentence.*

P1 Lines 23 – 28: what is the point you are trying to make by mentioning the acceleration in MSLR? Maybe it would be more helpful to look at total increases in SLR? This is what really will affect coastal areas. Are there more recent references?

*We mention especially the past acceleration of SLR, because the numbers illustrate in a striking way that changes are already going on. We revised this introductory text passage. We included numbers for total future increases in mean sea level rise and refer to the recently published IPCC Special Report on the Ocean and Cryosphere in a Changing Climate (IPCC 2019, 2019).*

P1 Line 28-29: worth including a reference and eluding to the reasons for this.

*We added references and modified the text passage in order to better explain the connections.*

P1 Line 28: highlight why changes in the tides are important – water level variations, extreme water levels, species distributions, changing currents, etc

*We revised this part of the manuscript.*

P1 Line 29 – P2 Line 7: are the changes you mention here due to changing tides or changing sea level? If the latter then they feel slightly out of context.

*We are referring to changes in tidal dynamics but these changes are directly related to changes in mean sea level. We modified the text passage to explain this more clearly.*

P2 Line 25: what sort of shelf models are you referring to here? Tide models? OGCMs?

*The shelf models meant are tide models. In the manuscript we included examples to clarify this aspect.*

P3 Shelf Model description(s): Given that your work is all about tides it would be helpful to include more information on the calculations of the tides in the model. How is bed roughness dealt with? Internal tides? Is this a 2D or 3D model? What happens at the open boundaries?

*The DSCMv6FM is a 2D model based on the shallow water equations that includes tide generating forces. Surge calculation is based on the inverse barotropic correction. At the open boundary 22 tidal constituents are used from GOT00.2 (Q1, O1, P1, K1, N2, M2, S2, K2, 2Q1, σ1, ρ1, χ1, π1, ϕ1, θ1, 2N2, μ2, v2, l2, L2 and T2) and 16 partial tides from FES2012 (Ssa, Mm, Mf, Msf, Mfm, S1, J1, M3, MNS2, R2, M4, MN4, MS4, S4, M6 and M8). Thus tides are produced/affected by two mechanisms, by astronomical factors (gravitational attraction between Earth, moon and sun, rotation of the Earth-moon and Earth-sun systems) and by non-linear processes (e.g. friction due to bed roughness) that affect the tidal constituents when propagating across the model domain.*
*The DCSMv6FM uses a spatially varying bed roughness. We modified the text passage to complete the model description.*

P3 Line 19: are you really using GOT00.2? The latest version is GOT4.7

*Yes, we are using GOT00.2. We believe that the data set is sufficient for our purpose, since this study is a case study and we are running both models with the same forcing. Thank you for the remark. We plan to update these data for future studies.*

P4 Line 5: why would you want sediments in your model?

*Since the numerical model UnTRIM² is principally able to compute the transport of suspended sediment, with this sentence we want to make clear, that for this study we have this process not included. In our study the focus lies on the response of tidal dynamics to sea level rise. The transport of suspended sediment is computationally intensive and not of primary relevance to tidal dynamics. To clarify this point, we included a short explanation.*

P4 Line 7: What parameters are used in the subgridscale option? How does this feed into the shallow water equations? Do both models have this option? How does turning this option on and off affect the results?

*Subgrid allows using coarser computational grids with high resolution data for bathymetry at subgrid level. The advantage of this technique is that large time steps can be used in the simulation due to the coarser computational grid. The algorithm that correctly represents the precise mass balance in regions where wetting and drying occurs and was derived by Casulli (2008) and Casulli and Stelling (2011). The computational grids are allowed to be wet, partially wet or dry. This implies that no drying threshold is needed. With the subgrid option the accuracy of the simulation results can be improved when using the same classical computational grid. Comparable results to simulations with a classical grid are obtained with using a coarser computational grid but with the subgrid technique. (Sehili et al., 2014)*

*For further information on the implementation of this method in the shallow water equation we refer to equation (7) in Sehili et al. (2014).*

*The DCSMv6FM uses the new flexible mesh capacities D-Flow-FM. The flexible mesh technique is another designation for the classical unstructured grid concept and thus, in contrast to the German Bight Model, does not include subgrid information.*

*We included the information on the subgrid option in the model description of the German Bight Model and clarified that DCSMv6FM does not use subgrid information.*

Table 1: It would helpful to include a root mean square error here as this gives an additional measure of the absolute model error rather than the relative error you get with the bias.

*We added the rmse and the bias of the M2 amplitude at all stations in the table 1.*

P6 Line 7: explain RMSE*?

*RMSE\* is the root-mean-squared-error normalised with the standard deviation.*
*We added this in the manuscript.*

P6: Given that you mainly discuss changes in M2 amplitude later on, it would be helpful to include an evaluation of the model performance in simulating the tides against the amplitudes at the stations and possibly also against a product such as TPXO or FES. This is probably more important than including an analysis of the water levels as it is not clear through which processes the water level errors arise (i.e. your model could perform very well at simulating tides but not for storms or vice versa).

*We chose a validation period in which wind velocities are small. Thus water levels are not influenced by storm surges. For this reason we believe that if the model simulates water level well it also simulates the tidal constituents (such as M2 amplitude) well. To complete the picture we, however, added the bias for the M2 amplitude between the measurements and the models at the stations in table 1.*
*In our opinion products such as FES or TPXO are not suited for validation purpose in our case. They are designed to model tides in the deep oceans. The bias increases on the shelf and especially at the coast where shallow water tides dominate. The error of these models on the coast is larger than the error we see in our validation. (https://www.aviso.altimetry.fr/en/data/products/auxiliary-products/global-tide-fes/description-fes2014.html)*

P7 line 12 – 13: This statement is not clear to me – why do you add SLR at the open boundary rather than over the whole domain? Is this the case for both models?
Can you explain in more detail how the model deals with flooding areas, i.e. the wetting and drying scheme? How does that work at low water and sea level rise? Surely, with SLR more normally dry areas are flooded and more energy is lost there?

*Water levels within the model domain are dominated by the forcing at the open boundary. Adding the mean sea level rise at the open boundary is a simple way to introduce sea level rise into the model domain. It would be more straightforward to add mean sea level also to the whole domain at the initial time step of the simulation period. However, the water signal introduced at the open boundary travels within a few days across the whole model domain. We chose the initialisation time of both models to be sufficiently long that a dynamical equilibrium is reached before analysing the data.*

*At the open boundary of the DCSMv6FM the mean sea level rise is added as a constant factor in the formula that calculates the tide using several harmonic constants (Zijl et al. (2013) eq. 1). At the open boundary of the German Bight Model water levels are used from DCSMv6FM which already include the effects of mean sea level rise on the shelf. At the corresponding text position in the manuscript we added this information.*

*Most numerical models control wetting and drying using thresholds for a minimum water depth. The models set the flow velocity to zero and take points out of the computational domain when a certain drying threshold is reached. Those points are "reactivated" when the water depth increases over the flooding threshold. The algorithm used in UnTRIM² is directly derived from the governing differential equation and is a substantial part of the numerical method. It guarantees mass conservation and allows any computational cell to be wet, partially wet or dry which is detected by having precisely zero water depth. No drying threshold is needed. For further information concerning the wetting and drying algorithm we recommend (Casulli, 2008) and the literature within. We have added the additional literature in the manuscript.*

*In principle rising mean sea level leads to a flooding of former dry area and thus more energy is dissipated on these areas. But it depends on the profile of the flooded area. If it is a steep profile then it could be that in the sum more or less the same amount of energy dissipates with or without MSLR, because the water depth increases (and dissipation decreases) in areas that were already flooded without mean sea level rise. When having a flat profile the area on which dissipation takes place increases more relative to the decrease of dissipation caused by deeper water levels. Thus no general statement is possible.*

P7 Simulations: How do you deal with open boundary tidal forcing with sea level rise? Harker et al. (2019) show that using present day tides at the model open boundaries leads to large changes in tidal responses on the Australian Shelf – can you comment on this for your work? Similarly, global studies such as Mawdsley et al. (2015) show global changes in water levels. Work by Wilmes et al. (2017) show global tidal responses to large-scale sea level changes on the levels of your 10 m simulation.
Do you allow for flooding in the large SLR scenario? What coastal defences do you assume?

*Thank you for this remark. Sea level rise will affect global tides. We, however, assume that the tidal constituents will not change at the open boundary of the shelf model DCSMv6FM. We add mean sea level rise as a constant value to the boundary conditions. For our study this is a reasonable assumption since it is designed as a conceptual study investigating the interaction between sea level rise and the representation of the bathymetry in the coastal zone. Our study does not characterise the future development of the tides. We clarified this point in the discussion.*

*Besides the uncertainty due to the assumption that the tidal constituents do not change at the boundary condition there are other uncertainties we did not account for. As mentioned in the paper one of the largest uncertainties is how the bathymetry near the coast will change due to MSLR. Due to mean sea level rise, e.g., a vertical growth of tidal flats is expected. Such changes in coastal bathymetry will affect tidal dynamics in the German Bight.*
*The wetting and drying algorithm is in principle still working when simulating large sea level rises. However, in the scenario with MSLR 10 m the model is flooded at all times, since the water cannot escape. The model domains of the models have a fixed vertical wall which is so high that it is not possible to flood the hinterland.*

Figure 4: Why does Emden have such a large error?

*Emden is located in the inner estuary of the Ems. Therefore it is a station which is principally harder to calculate in high accuracy. Especially with a model which has a relatively coarse resolution. We added the explanation also in the manuscript.*

Figure 3: what are the units? Explain mNHN in the caption

*mNHN denotes metres above the German datum which is a good approximation of mean sea level  We added an explanation in the caption.*

P10 Line 3: How do you calculate tidal amplitudes?

*Tidal amplitudes are estimated by a harmonic analysis of tides (Pansch, 1988), which is based on a Fourier decomposition of the water level time series into harmonic functions of pre-scribed tidal constituents. We added this explanation in the paper.*

P10 line 4-5: omit sentence

*Changed in the manuscript*

Figure 6. Which model performs better at the present-day tides?

*It is hard to decide which model performs better at present-day tides only from the picture since there is no spatial dataset of measurements. Both results seem plausible.*
*The comparison to measurements at individual gauges shows that both models calculate the tidal dynamics sufficiently accurate and the results of both models are in the same order of magnitude (see table 1). The DCSMv6FM performs slightly better in comparison which is probably a result of the calibration method (see 2.1). For the present day this calibration method provides good results but the question arises if this is also the case for future projections.*

P19 lines 4-5: '"The response to a MSLR of 10 m is more comparable in both models" – the differences between the two model setups are probably on a similar magnitude or larger as for the 0.8m SLR scenario, however, they are masked by the larger-scale differences occurring on the whole shelf area (whereas for a 0.8 m SLR the shelf-scale changes are much smaller). I would, however, argue that there are pretty big differences between Figure 6 e and f, especially around the islands in the southern part of the domain where the difference amounts to over 1 m in amplitude.

*Thank you very much for this comment. This is a very important aspect. You are right. We plotted the differences between Figure 6f and e and also between Figure 6d and c (supplement Figure S1). The differences in the region offshore of the North Frisian Wadden Sea are smaller but in a similar order of magnitude as in the case of MSLR 0.8 m. In the South the differences have opposite sign. That means, in contrast to MSLR 0.8 m, the response to MSLR 10 m in the shelf model is smaller than in the German Bight model.*
*We modified the section 3.1 accordingly.*

P19 lines 32-33: I'm not sure I agree – see comment above. The differences locally are pretty significant. Your large-scale, regional or shelf-wide responses are similar, but then arguably they are also similar (i.e. not very much happens) for small sea level increase.

*Following the argumentation of your comment above, we modified this statement.*

P20 line 24: "For higher mean sea level rise scenarios (10 m) the resolution of the bathymetry is less important" see comments above. I would conclude that, if looking at complex coastal areas such as the German Bight, highly resolved nearshore bathymetry is important for assessing the impact of sea level changes in these complex areas as the local responses can differ from the regional, offshore tidal changes. This is the case whether the forcing is large or small.

*After explicitly comparing the differences of the responses (see above) we agree. We deleted the statement in the conclusion. Thank you again for your careful reading and your remarks on this aspect.*

3 Technical comments
P2 Line 21: "1m" – space needed; also check remainder of the document
   *Changed in the manuscript*

P2 Line 23: defence -> defences
   *Changed in the manuscript*

P2 Line 28: After "Thus" add a comma
   *Changed in the manuscript*

P10 Lines 3 – 4: Figure 6 shows for both numerical models the M2 amplitude and its changes in response to mean sea level rise in the region of the German Bight.
-> Figure 6 shows the M2 amplitude and its changes in response to mean sea level rise in the region of the German Bight for both numerical models.

*Changed in the manuscript*

Table 2: You have quite a lot of different cases – remembering what case study 1 referred to later on in the manuscript is hard – why not label them something intuitive like GBM_ref_CS1 -> GBM_ref_NE and GBM_80_CS2 -> GBM_80_NE_CB

*Changed in the manuscript*

Figures 7 onwards: rather than having lots of single plot figures it would be better to condense your individual images into one or two figures with more subplots like you have done in Figure 6.

*Changed in the manuscript*

P13 lines 8 - 9: Similar to the shelf model the M2 amplitude increases in this case study in the German Bight. -> The M2 amplitude increases in the German Bight in this case study are now similar to the shelf model changes.

*Changed in the manuscript*

References

Casulli, V.: A high-resolution wetting and drying algorithm for free-surface hydrodynamics, Int. J. Numer. Meth. Fluids, 60, 391–408, https://doi.org/10.1002/fld.1896, 2008.

Casulli, V. and Stelling, G. S.: Semi-implicit subgrid modelling of three-dimensional free-surface flows, Int. J. Numer. Meth. Fluids, 67, 441–449, https://doi.org/10.1002/fld.2361, 2011.

IPCC 2019: IPCC Special Report on the Ocean and Cryosphere in a Changing Climate, in press, 2019.

Pansch, E.: Harmonische Analyse von Gezeiten- und Gezeitenstrombeobachtungen im Deutschen Hydrographischen Institut, Deutsches Hydrographisches Institut, Hamburg, Wissenschaftlich-Technische Berichte 1988-1, 2350, 1988.

Sehili, A., Lang, G., and Lippert, C.: High-resolution subgrid models: Background, grid generation, and implementation, Ocean Dynamics, 64, 519–535, https://doi.org/10.1007/s10236-014-0693-x, 2014.

Zijl, F., Verlaan, M., and Gerritsen, H.: Improved water-level forecasting for the Northwest European Shelf and North Sea through direct modelling of tide, surge and non-linear interaction, Ocean Dynamics, 63, 823–847, https://doi.org/10.1007/s10236-013-0624-2, 2013.

---

## Author Comment (AC2) · 8 Nov 2019

We thank the second reviewer for her/his effort on the review and for the helpful suggestions. In the following you find our answers and thoughts to your questions and remarks.

1. Methods and model description. The studies relies on two models. In the description of the models (sections 2.1 and 2.2), it seems that the simulations are done taking into account meteorological forcing, offshore salinity and river discharge. No information is provided on these conditions. In addition, if the regional model accounts for the salinity on its offshore boundary, we can guess that the river salinity is also accounted for.
Nothing is said, no value is given. Thus, the manuscript really required improvement on the description of the simulations (input conditions) and justification. Indeed, for the purpose of the study, why using 3D baroclinic simulations rather than 2DH simulations? Nothing is said.

> *Thank you for your careful reading. We added the missing information on the boundary and initial conditions in the model descriptions.*
> *3D baroclinic simulations allow taking into account stratification and variations in density which have a major influence on salt intrusion and distribution. We decided to include salt in the model since it is an important parameter for modelling tidal dynamics especially in the estuaries and near the mouths of the estuaries.*

2. Validation. First, the manuscript should provide the validation period over which they validate the tide. Second, even if they do not state it, the authors assume indirectly that the regional model (at high resolution) is better than the shelf model (at low resolution) to reproduce tide changes induced by sea-level rise. To support this assumption, a comparison of observed past tide change trends (using for instance literature results on tide gauges located in the study area) and results obtained simulating an additional and more moderate sea level rise scenario of e.g. 0.2 or 0.3 m, would be useful, with all the limits that such comparison has (additional mechanisms can contribute also modify the tide). But, as in Schindelegger et al. (2018), this would reinforce the paper.

> *We added the validation periods in section 2.3 Model validation..*
> *One of the main statements of the paper is that the shelf model (DCSMv6FM) and the regional model (GBM) show different responses to MSLR and that these different responses can be attributed to the different resolution of bathymetric information included in the models. Thus at least one of the two models does not simulate the correct response to MSLR. However, both models are likely incorrect. Especially, due to several uncertainties (e.g. missing morphodynamic processes) both model simulations are not able to predict the future response of the tidal dynamics to MSLR. Nevertheless, you are right, we assume that the regional model (with high resolution) simulates the response to MSLR more correctly than the shelf model (with low resolution) under the given boundary conditions (morphostatic simulations, high dikes at their current position). The assumption, that the regional model is more reliable, is based on the fact that physical processes are represented more accurately in the regional model (see e.g. Figures on current speed). We clearified this aspect at the end of the discussion section.*
> *A comparison of observed past tide change trends and model results with a moderate sea level rise scenario is difficult. In contrast to the model results observed past tide change trends include several factors that are not incorporated in the model simulations. Such factors are natural and anthropogenic morphodynamic changes as well as engineering measures (e.g. construction of embankments, construction of flood barriers). These factors, however, influence especially tide gauges closely located at the coast. Other challenges are vertical land motion and the natural variability caused by the meteorology in oberservational data. Also Schindelegger et al. (2018) found that the comparison of oberservational data and model results on the European Shelf is complicated by factors (e.g. dredging) not incorporated in the model simulations.*
> *In general, the comparison of model results with observed past tide change trends is an important basis for reliable future projections. In the light of the uncertainties mentioned we do not attempt to make future projections in this study. However, the comparison of model results*

*with observed past tide change trends in the German Bight is a challenging and important research question that should be pursued in further studies.*

3. Sensitivity of the tide changes to the bathymetric resolution. The manuscript would strongly benefit from a real sensitivity study of the tide changes to the bathymetric resolution, by investigating different bathymetric resolutions with the regional model, and not only the one corresponding to the shelf model. This would allow identifying if there is a bathymetric resolution below which there is no further improvement. Such result would allow the authors to make recommendations for the German Bight, and would strongly increase the impact of the work.

> *Thank you very much for this comment. We completely agree that an advanced sensitivity study would support further understanding of this topic.*
> *However, due to the unstructured computational grid that has different resolutions in different regions, the generation of computational grids with different resolutions is complex and not straightforward. Furthermore, the subgrid technology used in the regional German Bight model plays a crucial role. It allows to specify bathymetric details at a much higher resolution compared to the computational grid. With the subgrid option the accuracy of the simulation results can be improved when using the same classical computational grid. As shown in Sehili et al. (2014) different resolutions of the computational grid (within a certain range) do not influence the simulated results when using the subgrid option. Thus we suppose that a different resolution of the computational grid would not change the basic results. A sensitivity study investigating different grid resolutions and the role of bathymetric subgrid information, however, is an interesting research question for further studies.*
> *We added this idea and some thoughts on subgrid in the discussion section. In the model description of the German Bight Model more information on subgrid is given.*

References
Schindelegger M, Green JAM, Wilmes S-B, Haigh ID (2018) Can we model the effect of observed sea level rise on tides? J Geophys Res Oceans. https://doi.org/10.1029/2018J C0139 59

"On-line" Remarks
P1-Line 27: "flat" -> in most of the paper, the authors use "flat". I think that "low lying" is more relevant
> *We have changed it in this context. Later on we aim for the profile which is better described with "flat" in the Wadden Sea.*

P1-Line 27: add a reference to figure 1 and figure 2 (German Bight).
> *Changed in the manuscript*

P1-Line 29: "estuaries." -> reference?
> *We added a reference in the manuscript.*

P1-Line 29 -> P2, line 5: would better fit in the discussion section? Or remove it?

> *This text passage is part of the motivation why it is important to investigate how mean sea level rise influences tidal dynamics in the German Bight. We revised this part of the manuscript and hope that this intention is now clearer and that it now fits better to the rest of the introduction.*

Figure 4: hard to see the green star and blue points ! make a zoom for _RMSE=0 to 0.5
> *Figure added in the manuscript*

P15, line 5: make clear what is the mean current velocity. Is it the M2 - depth averaged current velocity? Is it the M2 depth averaged current velocity averaged over a given period (and is so, which period?)

*It is the depth averaged mean current speed analysed over a spring-neap-cycle in July 2010. We added this information in the manuscript.*

References

Casulli, V.: A high-resolution wetting and drying algorithm for free-surface hydrodynamics, Int. J. Numer. Meth. Fluids, 60, 391–408, https://doi.org/10.1002/fld.1896, 2008.

Casulli, V. and Stelling, G. S.: Semi-implicit subgrid modelling of three-dimensional free-surface flows, Int. J. Numer. Meth. Fluids, 67, 441–449, https://doi.org/10.1002/fld.2361, 2011.

Schindelegger, M., Green, J. A. M., Wilmes, S.-B., and Haigh, I. D.: Can We Model the Effect of Observed Sea Level Rise on Tides?, J. Geophys. Res. Oceans, 123, 4593–4609, https://doi.org/10.1029/2018JC013959, 2018.

Sehili, A., Lang, G., and Lippert, C.: High-resolution subgrid models: Background, grid generation, and implementation, Ocean Dynamics, 64, 519–535, https://doi.org/10.1007/s10236-014-0693-x, 2014.

Zijl, F., Verlaan, M., and Gerritsen, H.: Improved water-level forecasting for the Northwest European Shelf and North Sea through direct modelling of tide, surge and non-linear interaction, Ocean Dynamics, 63, 823–847, https://doi.org/10.1007/s10236-013-0624-2, 2013.

---

## Author Comment (AC3) · 8 Nov 2019

We thank the third reviewer very much for reviewing our paper and for her/his constructive advices to improve the manuscript. In the following we answered your questions and have indicated what information we added in our manuscript.

1. In the Introduction the sentence, "The German Bight located in the south-east of the North Sea with its flat coastal areas could be especially vulnerable" appeared a bit surprisingly. The authors started with a global perspective and I wondered why the German Bight was the area of choice. In addition, why is the German Bight especially vulnerable compared to other flat and low-lying coasts?

*We changed this part of the introduction. We focus on the German Bight as one example for low-lying coastal areas. The low-lying coast of the German Bight is not more vulnerable than other coasts with the same conditions like the Dutch coast. This part of the introduction shall say that the flat German coast with the Wadden Sea is more vulnerable than a steep rocky coastline like the east coast of Great Britain.*

2. As the authors motivated their study by differences in the response of global models, can the conclusions from the German Bight be generalized?

*Our motivation for this study is to show that depending on the model setup and especially the resolution of the bathymetry different responses to the same mean sea level rise can be estimated. This is especially effective in shallow coastal waters where the small-scale topographic structure has a great influence. Thus we would say that the conclusions could be transferred to other regions which show a comparable tidal regime and similar topographical characteristics like e.g. flat intertidal areas but could not be generalised without restrictions.*

3. Model set-up and experiments need some more explanation, in particular

a. Why did the authors choose to include meteorological forcing and not just used a tide-only simulation, in particular as they said they choose the summer to "ensure that the results are not influenced by storm surges or extraordinary high river discharge" (page 7).

*Tide-only simulations would have been also a good option. In the first place we decided to include meteorological forcing to be more realistic. Since we choose a summer period for our analyses, in which wind speeds are small, the meteorological forcing does not influence the results. Thus it is not critical whether we include meteorological forcing. We added a short explanation in the manuscript.*

b. The salinity boundary condition (page 5) needs some explanation as I expect that readers are not necessarily aware of what was done in the referenced project.

*We added additional information about the salinity boundary condition in the manuscript. The aim of the project AufMod was to develop a model-based tool to analyse long-term sediment transport and morphological processes. During this project a numerical model of the North Sea was set up. The used salinity boundary condition is a result of a reference simulation carried out during AufMod.*

c. A source for the river discharge data and their time resolution should be provided.

*The data for the river discharge is provided by the Water and Shipping Authorities and the "Gewässerkundliches Jahrbuch" which are published by the Hamburg Port Authority for the Elbe and the NLWKN for Ems and Weser. The time resolution of the data is one day. We added this information in the manuscript.*

4. Figure 6 and following: The gray color bar should be explained in the caption. I was not able to clearly identify gray areas in the Figures.

*We removed the grey colour bar in the figures where it isn't necessary. The grey colour which indicates dry areas is now explained in the caption.*

5. Page 19, line 29: What exactly is "sufficiently fine"? There are probably substantial changes in bathymetry over time as well. What would the authors then consider a "sufficiently fine" resolution?

*You are right; the formulation "sufficiently fine" is not a very clear choice. The answer to the question what we would consider as a "sufficiently fine" resolution cannot be given easily. It depends on the concrete research question.* In particular, for questions related to locations close to the coast the model bathymetry should incorporate the main characteristics of the Wadden Sea. *To give a proper answer further research is needed.* One way to explore which resolution is needed could be a sensitivity study, in which the resolution is systematically varied as suggested by reviewer #2. We added this aspect in the discussion at the corresponding text position.

---

## Author Comment (AC4) · 8 Nov 2019

Dear Philip Woodworth,

Thank you very much for your careful reading and your valuable hints and suggestions!

We fully agree that some figures required correction. We are grateful for your remarks in this matter. We have integrated your suggestions and comments into our manuscript.

p1, line 8 (and other places) - is the German Bight really in the SE of the North Sea rather than the east.

> *In our comprehension the German Bight is located rather in the south-east of the North Sea than in the east. That is certainly a matter of opinion. Also other authors describe the location of the German Bight in the south-east (e.g. Wahl et al., 2011; Albrecht and Weisse, 2012).*

figures 1 and 2. These have several errors. Fig 1 has degrees longitude and latitude swapped. But anyway they should read North Latitude (deg) and East Longitude (deg). In figure 1 deep water bathymetry, shown by dark blue, is a negative number whereas in Figure 2 it is a positive number. In Figure 2 it is hard to read the black names.

> *We corrected the figures and updated them in the manuscript.*

Explain somewhere that NHN is the German datum which is a good approximation of MSL.

> *An explanation is now given on page 6.*

Table 1. replace commas in the numbers with dots which is more normal internationally.

> *Changed in the manuscript.*

Figure 3. define MEZ in terms of UT
It is not necessary to have 00:00 in the times.

> *Changed in the manuscript.*

Figure 3. Why is there a slightly different set of 7 stations used for the 2 models (because of the grids presumably). explain better. It is not easy to spot the grouping of blue A,B etc. when printed - I had to blow up the pdf to see that. Anyway Target-Diagram would be better as 'Target Diagram' and things would be clearer if the maximum radius was 0.6.

> *The seven stations that we show in the diagram are the same spots in each model. The only difference is the order of the stations in the legend. I am afraid we could not change the order because it is hardcoded in the analysis and the plotting program.*
> *We zoomed in so hopefully the locations can be better seen now.*

p8, 5 and elsewhere - oscillation volume sounds odd to me although I am struggling to think of something else. perhaps have this sentence read: Therefore, the only difference between models concerns the volume of water exchanged through the tidal cycle, which we call the oscillation volume.

*We agree with you that "oscillation volume" (page 8, line 5) may lead to misunderstandings. The removed estuaries shorten the length of the estuary and inhibit the tidal wave from propagating further. The wave is reflected much earlier and therefore the oscillation behaviour changes in the corresponding areas.*
*In this context we don't mean the tidal prism thus the water volume that is exchanged through every tidal cycle. We mean the varied volume of the tidal basin itself. To make this clear we changed the manuscript.*

Figure 5. why does this have a different colour scale to Figure 2? It covers almost the same area.

*In figure 2 and also in figure 1 we show the whole model domain and the used colours should give information about the present depths. In figure 5 we use a different colour scale which looks more realistically. This colour scale gives the viewer a better idea of the tidal flats, the channels and the complex structure of the Wadden Sea (right hand side). The simplifications resulting from the coarser topography (left hand side) can then be better estimated.*

section 2.5 header. Please do not be so cryptic. Perhaps Analysis of Model Simulations. section 3 Model results

*Changed in the manuscript.*

p10, 26 - this is not surprising as the volume of water in the estuaries is small.

*We agree, but we were not sure before we conducted this study.*

Figure 6 and others. What is the second b/w scale for? Is that because of the wet/dry areas or what? Anyway it is not explained in the captions. It seems to me it could be just left off.

*We removed the grey colour bar in the figures where it isn't necessary. The grey colour which indicates dry areas is now explained in the caption.*

The paper does not discuss changes in phase lags, only amplitudes. Nothing to say about them?

*In our paper we focus on the amplitude of M2. For completeness, we added in the results section and in the supplement some information on the changes in the phase of M2.*
*Based on figure 6, figure S2 shows the phase of M2 in both models in the reference case and the respective changes by a rise of mean sea level of 0.8 m and 10 m. In both models the celerity of the tidal wave increases due to mean sea level rise. In the simulations with mean sea level of 10 m this increase is stronger than in the simulations with mean sea level of 0.8 m. This seems plausible since water depth increases.*

p15 current velocity should be current speed (velocity is a vector)

*We agree with you that current speed is the correct term. We changed the manuscript at the according positions.*

Figure 11 and 12 (a) remove the white arrow. You can't have a negative speed.

*Changed in the manuscript.*

Somewhere I noted R2 pointing to the recent Schindelegger et al. (2018) paper regarding model validation, and I was reminded of the Harker et al. (2019) paper in Ocean Science concerning the important aspect of whether model tides are allowed to change on an open boundary when MSL changes, and that should be made clear in the present paper.

*Thank you for this hint.*
*Contrary to the considerations from Harker et al. (2019) we assume that the tides will not change at the open boundary of the DCSMv6FM due to mean sea level rise since we are performing a case study. But nevertheless changes in tides on the European shelf due to shallow water effects are taken into account in the study since the boundary values of the German Bight Model are derived from the DCSMv6FM.*
*We added this additional information in the manuscript to make this more explicit.*

References

Albrecht, F. and Weisse, R.: Pressure effects on past regional sea level trends and variability in the German Bight, Ocean Dynamics, 62, 1169–1186, https://doi.org/10.1007/s10236-012-0557-1, 2012.

Harker, A., Green, J. A. M., Schindelegger, M., and Wilmes, S.-B.: The impact of sea-level rise on tidal characteristics around Australia, Ocean Sci., 15, 147–159, https://doi.org/10.5194/os-15-147-2019, 2019.

Wahl, T., Jensen, J., Frank, T., and Haigh, I. D.: Improved estimates of mean sea level changes in the German Bight over the last 166 years, Ocean Dynamics, 61, 701–715, https://doi.org/10.1007/s10236-011-0383-x, 2011.

---

## Author Response (AR2)

Dear Philip Woodworth,

Thank you very much for your remarks and the careful reading!

Concerning your comments to Table 1 and Figure 2 and 4, we would like to explain this in more detail.

The location based analysis program which we use at the BAW is very sensitive concerning data gaps in the measurements. In these situations the analysis aborted because the tool cannot detect all tidal high waters and tidal low waters. The calculation of the tidal characteristic numbers is not feasible at the locations of Emden and Büsum which likely fall dry at some time because the locations are in shallow areas like the Meldorfer Bight. Thus in these both cases it was not feasible to calculate a RMSE or a Bias for the thw, tlw and mtl.

In the "Target Diagram" synoptic data is used and no tidal characteristic numbers. That's why in this case all seven stations can be analysed. To create this "Target Diagram" the analysis is run for the DCSMv6FM and for the German Bight Model. The names of all locations are included in the simulation runs and the list of the location names is adopted in the analysis and is therefore fixed and cannot be changed. In the German Bight Model we have in total 118 locations where we get a model output. The DCSMv6FM does not include that many locations. Since the GBM has more locations, the order of the location names is different.

This circumstance is not pleasant and we can totally understand your comments, but we are not able to solve this problem in proper time. To fix this problem, all simulations would have to be re-run to change and harmonise the order of the locations in both models. In future studies this should be done at the beginning.

p1, 9 - drop 'By definition'. Maybe 'Models cannot adequately represent ..'

*Changed in the manuscript*

I did not understand this sentence as you show in Figure 6e,f that the 10 m pattern is much the same for the two models.

Maybe change to: .. amplitude increases in both models with largely similar spatial patterns. Anyhow this sentence needs changing as the end of the present one doesn't make sense.

*Changed in the manuscript*

- .. have been observed ... and 1990 at an average rate of 1.4 mm/year (IPCC, 2019).

This should be 1.4 and not 0.14.

*Changed in the manuscript*

.. by 2100

*Changed in the manuscript*

.. exceed several ..

then drop the IPCC, 2019 reference as given already.

    *Changed in the manuscript*

p2, 2 ... which are, in contrast ... coastlines, especially ..

    *Changed in the manuscript*

- .. in a changing climate .. in themselves ..

    *Changed in the manuscript*

- .. to the protection of coasts due to storm surges, but will also influence ..

    *Changed in the manuscript*

.. velocity will lead ...

*Changed in the manuscript*

- .. in Europe such as Rotterdam ..

    *Changed in the manuscript*

15 - measures,

    *Changed in the manuscript*

- dissipation along

    *Changed in the manuscript*

- dissipative areas

    *Changed in the manuscript*

- .. and realistic flood defence representations are needed ..

*Changed in the manuscript*

.. of flood defences but also the correct description of topography in shallow

    *Changed in the manuscript*

p3, 1 - to assess reliably

*Changed in the manuscript*

- ... MSLR. In particular, shallow ..

*Changed in the manuscript*

- structures might lead to imprecise [or incorrect?] results.

*Changed in the manuscript*

- equations.

*Changed in the manuscript*

19-21 This sentence is modelling jargon and makes no sense in English. Please reword.

*Changed in the manuscript*

- proper reference needed for the NOAA reference

*Changed in the manuscript*

- phases --> phase lags. And please check through the paper that phase is replaced by phase lag throughout.

.. of the 22 main tidal ..

drop (22). It is not clear what this is.

*Changed in the manuscript*

p4,1 - 16 --> Sixteen

*Changed in the manuscript*

2, and also in the reference list - Carrere has an accent over the first 'e'

*Changed in the manuscript*

- journals tend not to like footnotes so can you move the words of FN1 to this caption which is anyway where the reader first spots mNHN. I suggest

*Changed in the manuscript*

Figure 1: Domain of the DCSMv6FM model. The black box ... Bight. The unit ..

*Changed in the manuscript*

p5, 2 - equations

*Changed in the manuscript*

- occur

*Changed in the manuscript*

- .. boundary, water ..

*Changed in the manuscript*

- The salinity boundary condition employed is a result of a simulation ,,

*Changed in the manuscript*

- estuaries,

*Changed in the manuscript*

this sentence needs a reference adding.

*The reference for the river discharge is given in the next sentence.*

- in the estuaries

*Changed in the manuscript*

what does preceding mean here? preceding AufMod? In which case say 'by the KLIWAS project which preceded AufMod (Seiffert ...

*We meant that KLIWAS is the preceding project to the current one (Network of experts of the BMVI). But this is not that relevant and therefore we deleted this advice.*

reach --> range

*Changed in the manuscript*

of the water level, move the footnote as mentioned.

A sufficiently
*Changed in the manuscript*

- is included using the same ..
*Changed in the manuscript*

- drop the reference. It has already been given on the preceding page
*Changed in the manuscript*

p6, line 1 caption - The area within the black
*Changed in the manuscript*

- the error message is presumbly for Table 1 and there are other error messages in this document. Why was this not spotted?
*Changed in the manuscript. Sorry for this!*

and Table 1 - tidal mean water is usually called mean tide level
*Changed in the manuscript*

- listed for 5 different ..

There are 5 stations in Table 1 but 7 in Figure 2. Why are not all 7 in the table? Also the order in the table is not the same as in the figure: Helgoland and Hornum are swapped. Please tidy up these things.
*See comment above*

- what does 'measured data' mean? You mean to hourly (or whatever) water level measurements? Reword something like

... compared with hourly measurements of water level at seven stations located ...

*Changed in the manuscript*

16 - normalized by the

    *Changed in the manuscript*

p7, 1 - normalized by the

    *Changed in the manuscript*

line 1 Table 1 caption. tidal mean water should be mean tide level. Also the caption should say that the units in the table are metres.

    *Changed in the manuscript*

Figure 4. I mentioned last time that I didn't understand why, if the 7 stations are the same as you say, they have different names in the box and in a different order. Also they are in a different order to those in Figure 3. Also why does Figure 3 have 1-7 and Figure 4 has A-G. These should be the same. Also the caption should say the numbers are normalized ones. Anyway reorder:

.. for the comparison on the DCSMv6FM and German Bight model .. so
this sentence is the same order as the box.

Then, I really don't understand why we have to have a different order as
you say in the last sentence of the caption. Please remove that and tidy things up.

*See comment above*

p8, 16 ... DSCMv6FM and throughout its domain.

    *The MSLR is added only at the boundary of the DCSMv6FM model. For the initial water level a value of 0 mNHN is assumed. A sufficiently long initialisation time ensures that the model reaches a dynamical equilibrium before the*

*simulations start.*

- This appears to be a reasonable assumption ..

    *Changed in the manuscript*

- .. conceptual one to investigate the

*Changed in the manuscript*

p9, 1 - .. coastal zone, rather than fully characterising the future ..

*Changed in the manuscript*

- time series

*Changed in the manuscript*

3 - Sufficiently long ..

*Changed in the manuscript*

- reword (I think):

... GBM_80_NE), the main difference to the reference runs is that the volumes of the tidal basins are changed. In addition, no river discharge is included. A simulation of the original reference model (GBM_ref) but with no river discharge is denoted GBM_ref_noQ (Table 2).

*Changed in the manuscript*

p10, 9 - The results for wet areas ..

*Changed in the manuscript*

- please could you spell out the notation? e.g. the overbar on Fzero refers to the depth averaging. The suffix H means what? And to be clear say that U is the 2-dimensional depth-averaged velocity vector. Otherwise U will look to be a scalar to some readers.

*Changed in the manuscript*

Also I mentioned last time that I didn't think Kang was a good reference, rather than a standard text book, but if you want to have it then at least in the references give the web address which I think is https://searchworks.stanford.edu/view/8804623

*Changed in the manuscript*

p11, 10, 11 and 13 - supplement --> supplementary

*Changed in the manuscript*

- figure S2. I think you need an extra sentence here to say exactly what the figure is indicating.

Please see also below my comments on the colour scale you have used.

*Changed in the manuscript*

- .. removal of the estuaries in the German Bight Model. In .. considered and there is no river discharge in the two runs.

*Changed in the manuscript*

p12, 3 - spotted and the general

*Changed in the manuscript*

At this point I realised that you sometimes refer to places you are of course familiar with e.g. North Frisian Wadden Sea, Weser estuary etc. which many readers will not know. Could you please go through the text and where you mention places like this perhaps add a few extra words to say where they are? Ideally one would have a map with the names on but I won't insist on that.

*Added in figure 2*

p15, 5 - Figure 7a should be 8a

*Changed in the manuscript*

- restricted locally

*Changed in the manuscript*

- 7b should be 8b

*Changed in the manuscript*

- model response (Figure 6c).

*Changed in the manuscript*

- drop (Figure 6c) here

*Changed in the manuscript*

p16, 7 - .. DCSMv6FM (Figures 6c and 8b), we ..

.. dissipation rate in the different runs (Table 3).

*Changed in the manuscript*

p17, 2 - (shown by the polygon)

*Changed in the manuscript*

3-4 - I would drop the 2 last sentences. They have been explained before.

*Changed in the manuscript*

- In both situations with the highly ..

*Changed in the manuscript*

- drop this 'case study' business and just reword:

.. between the runs. With fine bathymetry, the dissipation .... whereas with coarse bathymetry ...

*Changed in the manuscript*

.. in the near-shore parts of the German Bight.

*Changed in the manuscript*

- as I mentioned last time I would drop (global). This is not global! It is the German Bight. Instead say: In the domain-averaged . which I think is what you mean.

*Changed in the manuscript*

p18, 5 - current --> currents. andd drop velocities

*Changed in the manuscript*

Table 4, line 2 - averaged over the entire German Bight Model domain ..

and drop (global) on line 3. And in the table itself, column 4 say domain and not global

*Changed in the manuscript*

p21, 3 - 10 min --> 10 m in

*Changed in the manuscript*

- in two case studies [why do you now start talking about steps?]

*Changed in the manuscript*

- the first study

*Changed in the manuscript*

- the second study

*Changed in the manuscript*

- Within this second study, the German Bight Model was found to respond in

*Changed in the manuscript*

- in a way

*Changed in the manuscript*

14-16 - I think this needs rewording. It says first that overflowing is not allowed then it says it is. Please read and make clear.

*Changed in the manuscript*

- 'in the same characteristic' --> 'to the same extent'

*Changed in the manuscript*

p22, 6 - using subgrids.

*Changed in the manuscript*

- characterised by larger intertidal flat areas relative to channel areas

*Changed in the manuscript*

- I would start this paragraph as follows:

Several points can be mentioned concerning the extent to which model simulations such as described in this paper can be applied to estimate the tidal response to MSLR in a real future.

*Changed in the manuscript*

- drop 'concerning the point of matching reality'

*Changed in the manuscript*

- ... rise. For example, a vertical growth of tidal flats can be expected given a significant MSLR (Hofstede

*Changed in the manuscript*

- the simulations with 10 m MSLR especially

*Changed in the manuscript*

32 - These simulations are included here only to ... system response ..

*Changed in the manuscript*

p23, 2 - .. Bight, the small-scale representation of the bathymetry .. role in the ..

*Changed in the manuscript*

- and the geographic area of interest.

*Changed in the manuscript*

- .. the response of the wider North Sea, the use of a

*Changed in the manuscript*

References - Carrere (see above), IPCC 2019 update the reference, Kang (see above)

*Changed in the manuscript*

Figure S2. phase should be phase lag. The colour scales for c,d,e,f are ok but that for a,b is not. 0 and 360 degrees is the same angle and so should be the same colour, so this needs to be a rotating colour scale. Please look at any paper on tides that shows phase lags.

*Changed in the manuscript*

[revised manuscript text omitted]

---

## Author Response (AR3)

Dear Phil Woodworth,

Thank you very much for accepting my paper. I am very happy about that!

5   p17, line 16 - change 'minutely' to '1-minute'.
        *Changed in the manuscript*

    p22, line 28 - ... shows the phase lags of M2 in the German Bight Model and in the DCSMv6FM and their corresponding
    changes due ..
10      *Changed in the manuscript*

    p34, line 1 - remove the first 'especially'
        *Changed in the manuscript*

15  SM Figure S1b - Thanks for changing to the rotating colour scale for (a) and (b), but don't you think you need a rotating
    scale also for (e) and (f) as there are big areas of black off the limited scale you use? I'm not sure about (c) and (d). See what
    you think.

        *I don't think that SM Figure S2 c), d), e) and f) need a rotating colour scale. Below you find a figure which shows*
20      *an extract of figure S2c). In this plot I have added the exact difference amount at the grid elements. The large*
        *numbers in the area you mentioned, result from the jump from 360 to 0 degree which takes place in this area. Phase*
        *lag of model A = 2 degrees, phase lag of model B = 360 degrees, difference B-A = 358 degrees. In these cases it*
        *would not matter which colour scale is used, the jump is also present in a rotating scale. That's why I would not*
        *change the colour scale in these figures.*

[revised manuscript text omitted]